# Optimal estimation retrieval of tropospheric ammonia from the Geostationary Interferometric Infrared Sounder onboard FengYun-4B

Zhao-Cheng Zeng[1,*], Lu Lee[2], Chengli Qi[2], Lieven Clarisse[3], and Martin Van Damme[3,4]

[1]School of Earth and Space Sciences, Peking University, Beijing 100871, China
[2]Innovation Center for FengYun Meteorological Satellite, Key Laboratory of Radiometric Calibration and Validation for Environmental Satellites, National Satellite Meteorological Center, China Meteorological Administration, Beijing 100081, China
[3]Université libre de Bruxelles (ULB), Spectroscopy, Quantum Chemistry and Atmospheric Remote Sensing, Brussels, Belgium
[4]Royal Belgian Institute for Space Aeronomy, Brussels, Belgium

*Correspondence to: Z.-C. Zeng (zczeng@pku.edu.cn)

**Abstract.** Atmospheric ammonia ($NH_3$) is a reactive nitrogen compound that pollutes our environment and threatens public health. Monitoring the spatial and temporal variations is important for quantifying its emissions and depositions and evaluating the strategies for managing anthropogenic sources of $NH_3$. In this study, we present an $NH_3$ retrieval algorithm based on the optimal estimation method for the Geostationary Interferometric Infrared Sounder (GIIRS) onboard China's FengYun-4B satellite (FY-4B/GIIRS). In particular, we examine the information content based on the degree of freedom for signal (DOFS) in retrieving the diurnal $NH_3$ in East Asia, with a focus on two source regions including North China Plain and North India. Our retrieval is based on the FengYun Geostationary satellite Atmospheric Infrared Retrieval (FY-GeoAIR) algorithm and exploits the strong $NH_3$ absorption window 955-975 $cm^{-1}$. Retrieval results using FY-4B/GIIRS spectra from July to December 2022 show that the DOFS for the majority ranges from 0 to 1.0, mainly depending on the thermal contrast (TC) defined as the temperature difference between the surface and the lowest atmospheric layer. Consistent with retrievals from low-earth-orbit (LEO) infrared sounders, the detection sensitivity, as quantified by the averaging kernel (AK) matrix, peaks in the lowest 2 km atmospheric layers. The DOFS and TC are highly correlated, resulting in a typical "butterfly" shape. That is, the DOFS increases when TC becomes either more positive or more negative. The $NH_3$ columns from FY-4B/GIIRS exhibit significant diurnal cycles that are consistent with the day-night gradient from the collocated IASI retrievals in North China Plain and North India for the averages in July-August, September-October and November-December, respectively. Collocated point-by-point intercomparison with the IASI $NH_3$ dataset shows generally good agreement with a small systematic difference in the summer months that may be attributed to the slight difference in a priori profiles. This study demonstrates the capability of FY-4B/GIIRS in capturing the diurnal $NH_3$ changes in East Asia, which will have the potential to improve regional and global air quality and climate research.

## 1. Introduction

Atmospheric ammonia (NH₃) is a reactive nitrogen compound that plays an important role in the global nitrogen cycle (**Galloway et al. 2004**). Its emissions to the atmosphere are destined to increase in the coming decades primarily driven by agriculture emissions coming from the excess use of nitrogen fertilizers (**Fowler et al., 2013**). Changes in NH₃ emissions have many negative environmental impacts, including the loss of biodiversity (**Erisman et al., 2013**), the eutrophication of water bodies and the acidification of terrestrial ecosystems (**Paerl et al., 2014**), and the change in radiative forcing that affect global and local climate

(**Abbatt et al., 2006; Isaksen et al., 2009**). In addition, ammonia reacts with acids (e.g., $H_2SO_4$, $HNO_3$) and produces ammonium containing aerosols that degrade air quality (**Seinfeld and Pandis, 2006**). Monitoring the spatial distribution and temporal variations of atmospheric NH₃ is therefore important for quantifying NH₃'s emissions and depositions and evaluating the strategies for managing anthropogenic sources of NH₃.

Over the past decade, space-borne observations of NH₃ have provided measurements with daily global coverage that greatly

improve our understanding of the sources, depositions, and variabilities of NH₃ (e.g., **Zhu et al., 2015; Warner et al., 2017; Van Damme et al., 2021**). The capability of using an infrared hyperspectral sounder to detect NH₃ was first demonstrated using observations from the Tropospheric Emission Spectrometer (TES; **Beer et al., 2008; Shephard et al., 2011**) and the Infrared Atmospheric Sounding Interferometer (IASI; **Clarisse et al., 2009, 2010; Coheur et al., 2009**). In addition, NH₃ has been detected in the Asian summer monsoon upper troposphere using the emission spectra from the infrared limb sounder Michelson

Interferometer for Passive Atmospheric Sounding (**MIPAS; Höpfner et al., 2016**). Currently, long-term measurements of NH₃ are available from IASI (**Van Damme et al., 2021**), the Cross-track Infrared Sounder (CrIS; **Shephard et al., 2020**), the Atmospheric Infrared Sounder **(**AIRS; **Warner et al., 2016**), and the Thermal and Near-infrared Spectrometer for Observation-Fourier Transform Spectrometer (TANSO- FTS; **Someya et al., 2020**). However, the above-mentioned polar-orbiting satellites can only make up to two overpass measurements each day over the same location. The diurnal cycle of NH₃ is therefore under-

constrained from polar-orbit observations. Since NH₃ is a highly reactive compound, it has a relatively short lifetime ranging from a few hours to days (**Aneja et al., 2001**). As a result, NH₃ presents a large spatial heterogeneity and temporal variability. The important information on the diurnal cycle is therefore critical to constrain the emission, deposition, and transport processes of NH₃ and the role of meteorological conditions in driving these processes.

The Geostationary Interferometric Infrared Sounder (GIIRS) onboard the FengYun-4B satellite was launched in 2021 with

improved sensitivity over its predecessor FY-4A/GIIRS launched in 2016 (**Yang et al., 2017**). GIIRS was designed to probe the three-dimension water vapor and temperature profiles for weather forecast purposes. With its high spectral resolution (0.625 cm⁻¹) and sensitivity comparable to current low-earth-orbit (LEO) satellites, FY-4B/GIIRS is suited for detecting the changes of various atmospheric trace gases (e.g., **Zeng et al., 2022**). The strength of using FY-4B/GIIRS lies in its capability to scan East Asia every two hours with a spatial resolution of 12 km, offering a unique opportunity to constrain the diurnal cycles of critical atmospheric

composition at high spatial resolution. The application of GIIRS in detecting NH₃ has been successfully demonstrated using spectra collected by FY-4A based on an IASI retrieval method (**Clarisse et al., 2021**). The study showed that the unprecedented temporal sampling of GIIRS enables the measurement of diurnal and nocturnal variations of NH₃. Since the retrieval method used in **Clarisse et al. (2021)** is based on hyperspectral radiance index (HRI) and a trained neural network that relates HRI to NH₃ columns (**Whitburn et al., 2016**), information content analysis for the GIIRS spectra was not fully investigated.

Optimal estimation method (**Rodgers, 2020**) that enables information content analysis has been applied in previous studies on retrieving NH₃ from space (**Clarisse et al., 2009, 2010; Shephard et al., 2011; Warner et al., 2016, Shephard et al., 2011,2015; Someya et al., 2020**). It was found that the information available from the infrared sounder spectra for quantifying NH₃, especially its abundance in the PBL, strongly depends on the thermal contrast (TC), which is defined as the temperature difference between

the surface skin and the lowest atmospheric layer. Because of its short lifetime, $NH_3$ is mostly concentrated in the planetary boundary layer (PBL) and the peak sensitivity of $NH_3$ detection from an infrared sounder is also found to be close to the surface. Previous studies (e.g., **Clarisse et al., 2010**) concluded that the TC and its abundance are two important factors that determine the intensity of the $NH_3$ spectral signature, and hence the $NH_3$ information content. In general, higher $NH_3$ concentrations and larger TC result in a more accurate estimate of $NH_3$ from space-borne measurements.

In this study, we applied the FY-GeoAIR retrieval algorithm developed by **Zeng et al. (2022)**, based on the optimal estimation theory to retrieve $NH_3$ from FY-4B/GIIRS. The primary goals are to quantify the information content of FY-4B/GIIRS observations in constraining the diurnal cycle of $NH_3$ columns and to assess the impact of TC on the retrieval accuracy. The retrieval algorithm uses the absorption features of $NH_3$'s $v_2$ rotational-vibrational band centered around 10.5 μm (~950 cm$^{-1}$). Our retrieval is implemented using the absorption micro-window (955-975 cm$^{-1}$) that contains the strongest $NH_3$ absorption feature. A similar micro-window has been used to retrieve $NH_3$ using IASI (**Clarisse et al., 2010**) and GOSAT (**Someya et al., 2020**), while neural-network-based studies using hyperspectral radiance index (HRI) adopted a much wider window (e.g., **Whitburn et al., 2016**). The remainder of the paper is organized as follows. The FY-4B/GIIRS observation mode and the collected spectra are introduced in **Section 2**; In **Section 3**, the FY-GeoAIR algorithm for $NH_3$ retrieval is described; The retrieval results, information content analysis, intercomparison with IASI $NH_3$ dataset, and related discussion are presented in **Section 4**, followed by conclusions in **Section 5**.

## 2. GIIRS onboard FengYun-4B

FY-4B/GIIRS is an infrared Fourier transform spectrometer based on a Michelson interferometer located at an altitude of 35,786 km above the equator at 133°E. The primary goal is to probe the three-dimensional atmospheric structure of temperature and water vapor over east Asia for improving numerical weather forecast. **Figure 1(a)** shows the coverage of FY-4B/GIIRS over East Asia and part of South Asia and Southeast Asia in one measurement cycle which lasts 2-hour. In each cycle, GIIRS makes 12 horizontal scans from north to south. Each scan sequence consists of 27 fields-of-regards (FORs) from west to east that collects upwelling infrared radiation of Earth scenes (ESs), followed by one deep space (DS) and one internal calibration target (ICT) measurement for ES radiometric calibration. For each FOR, a 2-dimension infrared plane array detector, containing 16×8 pixels with a sparse arrangement, conducts the measurement over the target region. The starting hours for the 12 measurement cycles in a day were 0, 2, 4, …, 22h UTC, respectively, and have been changed to 1, 3, 5, …, 23h UTC, respectively, after September 06, 2022. The observation domain, as shown in **Figure 1(a)**, covers the two important $NH_3$ emission source regions in Asia: The North China Plain and North India, as indicated by the bottom-up inventory map in **Figure 1(b)**. The FY-4B/GIIRS observed spectra include a long-wave infrared band from 680 to 1130 cm$^{-1}$ and a mid-wave infrared band from 1650 to 2250 cm$^{-1}$ with a uniform spectral resolution of 0.625 cm$^{-1}$. With low instrument noise and a high spectral resolution similar to current LEO infrared sounders, GIIRS is in principle capable of measuring trace gases, including $NH_3$ and carbon monoxide (**Zeng et al., 2022**), and providing full day-night diurnal cycle observations. The spatial footprint size of each pixel on the Earth's surface is about 12 km at Nadir, which is an improvement over FY-4A/GIIRS (16 km). **Figure 1(c)** shows an example of GIIRS spectra in the long-wave band. Post-launch assessment of the radiometric performances of FY-4B/GIIRS using a series of blackbody calibration experiments showed that the noise equivalent differential radiance (NedR) on average in the long-wave infrared 900-1000 cm$^{-1}$ bands, covering the $NH_3$ absorption channel, is about 0.1 mW/(m$^2$·sr·cm$^{-1}$). The corresponding noise equivalent differential temperature (NedT) on average is about 0.1 K@280K blackbody. The low instrument noise for FY-4B/GIIRS, comparable to existing infrared sounders (e.g., ~0.2K@280K for IASI and ~0.04K@280K for CrIS; **Van Damme et al., 2014; Shephard et al., 2020**), makes it possible to accurately retrieve $NH_3$ over East Asia.

All cloud-screened FY-4B/GIIRS spectra acquired over land with a viewing zenith angle less than 70° are used in the retrieval. To filter out cloudy pixels, we use the higher-resolution (4 km) level-2 cloud mask (CLM) data product from the Advanced Geostationary Radiation Imager (AGRI) onboard FY-4B. When at least 7 of the 9 collocating AGRI pixels are either clear or probably clear, the GIIRS pixel is labeled as clear.

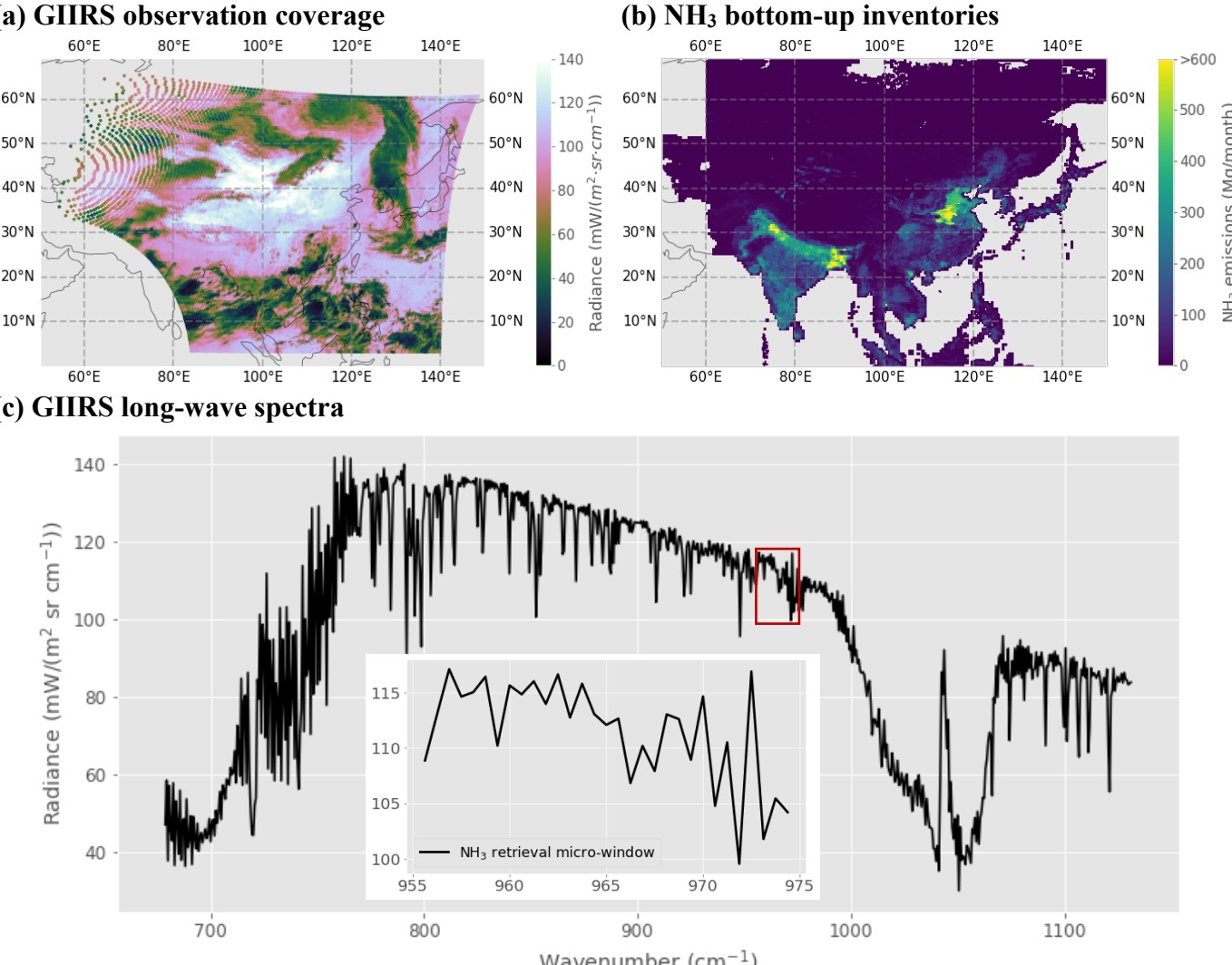

**(a) GIIRS observation coverage**

**(b) NH₃ bottom-up inventories**

**(c) GIIRS long-wave spectra**

**Figure 1. (a) GIIRS observation coverage color-shaded using the observed radiance at 900 cm⁻¹ from FY-4B/GIIRS measurements at 12-13h (Beijing Time) on July 07, 2022; (b) Bottom-up NH₃ inventory emissions averaged from July to December of 2010. These are the total amount combining emissions from agricultural, industrial, power plant, residential, and transportation sectors. This emission inventory is adopted from MIX dataset (Li et al., 2017); (c) Example of FY-4B/GIIRS measured long-wave spectra, which covers the NH₃ absorption band centered around 950 cm⁻¹. The micro-window for NH₃ retrieval is also shown in red rectangle and as inset. The NH₃ absorption features are shown in Figure 2.**

**3. The FY-GeoAIR retrieval algorithm for NH₃**

The FengYun Geostationary satellite Atmospheric Infrared Retrieval (FY-GeoAIR) algorithm was originally developed for retrieving carbon monoxide from FY-4B/GIIRS spectra. The algorithm combines a forward radiative transfer model to simulate upwelling thermal radiation and an optimal estimation-based inverse model to retrieve trace gases and auxiliary parameters from the observed spectra. Here, a brief introduction to FY-GeoAIR is given with descriptions of changes needed to adapt it to NH₃ retrieval. More details about the FY-GeoAIR retrieval algorithm can be found in **Zeng et al. (2022)**.

### 3.1 The Forward radiative transfer model for simulating observed spectra

An accurate radiative transfer (RT) model is an important component in the inversion system for simulating upwelling thermal radiation that would be observed by FY-4B/GIIRS when given the relevant atmospheric, surface, and instrumental parameters as inputs. The upwelling spectral radiance is computed following the radiative transfer theory for thermal radiation that has been described in **Clough et al. (2006)** and **Hurtmans et al. (2012)**. Under clear sky conditions, scattering by clouds and aerosols can be ignored. The upwelling radiance received by FY-4B/GIIRS can be accurately approximated by the sum of four main components,

including the upwelling surface emission, the upwelling atmospheric emission integrated from the bottom- to the top of the atmosphere, the surface-reflected downwelling atmospheric emission, and the surface-reflected solar radiation. Although the last component is very small in the long-wave band, it is added here for completeness. The absorption micro-window (955-975 $cm^{-1}$) used for retrieval is shown in **Figure 2(a)**. The micro-window contains strong $NH_3$ absorption features that are distinguishable from important interference gases ($CO_2$, $O_3$ and $H_2O$) in the absorption window, as demonstrated in **Figure 2** from a sensitivity

experiment that compares absorptions of $NH_3$ and perturbed interference gases.

The atmospheric, surface and instrumental parameters used to drive the forward model are adopted from various sources. These parameters are the atmospheric state such as the profiles of temperature, water vapor, and atmospheric composition, the surface parameters such as the surface emissivity, and the instrumental specifications such as instrument spectral response function and observing geometries. Specifically, the atmospheric temperature, $H_2O$, and $O_3$ data are extracted from European Centre for

Medium-Range Weather Forecasts (ECMWF) Reanalysis v5 (ERA5) reanalysis (**Hersbach et al., 2020**); $CO_2$ is extracted from ECMWF Copernicus Atmosphere Monitoring Service (CAMS) global inversion-optimized greenhouse gas fluxes and concentrations (**ECMWF, 2022**); For the surface land data, we used the global infrared land surface emissivity database from the University of Wisconsin-Madison (UOW-M) (**Seemann et al., 2007**). The emissivity values at 925 $cm^{-1}$ and 1075 $cm^{-1}$ are used to estimate the a priori emissivity for the retrieval micro-window. Two factors (slope and curvature) are used in the state vector to

scale the wavelength dependent emissivity values; Surface skin temperature and surface pressure are extracted from ERA5 hourly data on single level (**Hersbach et al., 2020**). The absorption coefficient look-up tables for calculating gas absorption are built using the extensively validated Line-By-Line Radiative Transfer Model (LBLRTM v12.11; **Clough et al., 2005**).

Since the $NH_3$ is short-lived and highly concentrated in the PBL, we therefore only retrieve the layers below 200 hPa. The forward model uses fixed vertical grids with equally separated layers with similar thickness (about 1 km for layers below 200 hPa

and about 5 km for layers above), which is close to the grid settings in **Hurtmans et al. (2012)** and **Clough et al. (2005)**. The thickness of the bottom layer is variable and determined by the surface pressure of a specific location. The number of layers below 200 hPa ranges from 7 (for high altitude regions such as the Tibet Plateau) to 11 layers (for low altitude regions such as the ocean).


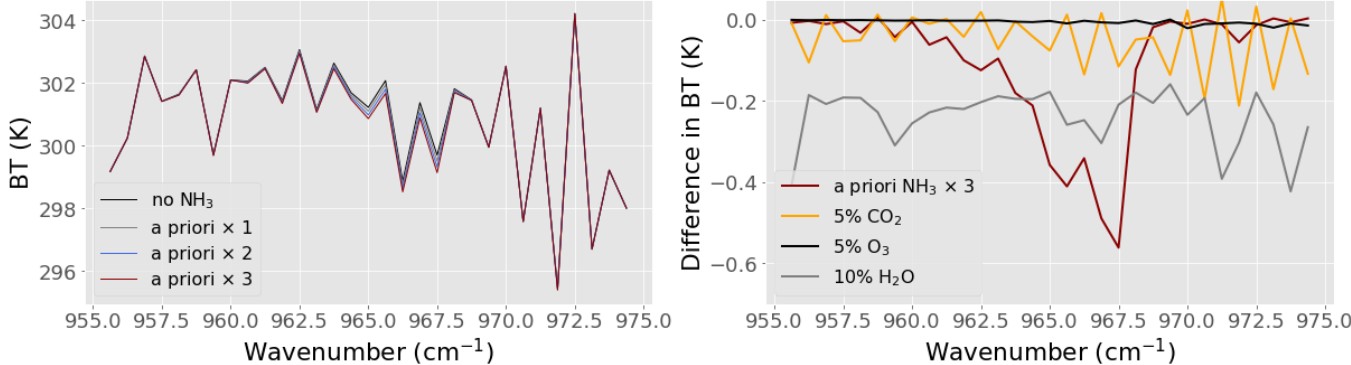

**Figure 2. Sensitivity experiment comparing absorptions of NH₃, CO₂, O₃ and H₂O in the 955-975 cm⁻¹ micro-window used for FY-4B/GIIRS NH₃ retrieval. The micro-window covers the strong absorption features of the NH₃ $v_2$ vibrational-rotational band. (Left) the comparison of simulated spectra using the forward model with three different NH₃ concentrations based on the a priori NH₃ profiles shown in Figure 3; (Right) the absorption difference between NH₃ and important interference gases including CO₂, O₃ and H₂O that have been perturbed by a certain factor.**

**Table 1. Parameters in the state vector to be retrieved from the retrieval algorithm.**

| variables | no. of variables | a priori values | a priori uncertainty | descriptions |
|---|---|---|---|---|
| NH₃ | 11 | fixed | see **Figure 3** | derived from GEOS-CF simulations |
| H₂O | 11 | ECMWF ERA5 reanalysis | 30% | |
| Interfering trace gases (CO₂, O₃, CFC12, HNO₃) | 5 (in total) | CO₂ (ECMWF CAMS); O₃ (ECMWF ERA5); CFC12 (LBLRTM standard profile); HNO₃ (LBLRTM standard profile). | 5% | only profile scaling factors are retrieved; a uniform 5% uncertainty is assumed for these minor trace gases that have very weak absorption features |
| Surface skin temperature | 1 | ECMWF ERA5 | 5K | |
| Air temperature profile | 1 | ECMWF ERA5 | 0.5% | only a profile scaling factor is retrieved |
| Surface emissivity slope and curvature | 2 | [0.0, 0.0] | [0.1%, 0.01%] | The a priori surface emissivity data come from University of Wisconsin-Madison (**Seemann et al., 2007**) |

### 3.2 The Retrieval algorithm in FY-GeoAIR based on optimal estimation theory

The goal of the retrieval algorithm for retrieving NH₃ from FY-4B/GIIRS based on optimal estimation theory is to find a solution for the state vector, which consists of NH₃ profile and auxiliary parameters, such that the simulated spectra from the RT forward model best fit the measured spectra. The auxiliary parameters include H₂O profile, scale factors for the columns of the remaining interference gases, surface skin temperature, and scale factor for the atmospheric temperature profile. The solution from the retrieval algorithm is the state vector which minimizes the spectral fitting error. The Levenberg-Marquardt modification of the Gauss-Newton method is used to search the solution. The optimal estimation method has been described thoroughly in **Rodgers (2000)** and applied in several previous studies by the group (**Zeng et al., 2017; Zeng et al., 2021; Natraj et al., 2022; Zeng et al., 2022**). The parameters in the state vector to be retrieved from the algorithm are listed in **Table 1**. Vertical profiles of NH₃ and H₂O are retrieved, while for minor interference gases, total columns are retrieved by scaling an a priori profile. Other parameters to be retrieved include the surface skin temperature, a scaling factor for the atmospheric temperature profile, and the slope and curve for the surface emissivity. We only retrieve the layers below 200 hPa and use the a priori for layers above to compute total columns.

Two important metrics from optical estimation method for interpreting the retrieval results are the DOFS and Averaging Kernel (AK) matrix. AK matrix is a metric that quantifies the sensitivity of the retrieval to the true state by the observing system. The full AK matrix ($m \times m$) is given by:

$$A = (K^T S_\varepsilon^{-1} K + S_a^{-1})^{-1} K^T S_\varepsilon^{-1} K \qquad (1)$$

where each element $A_{ij}$ of A represents the derivative of the $NH_3$ retrieval at level $i$ with respect to the $NH_3$ truth at level $j$; The matrix dimension $m$ is the number of atmospheric layers; $K$ is the Jacobian matrix, which is the first derivative of the forward model with respect to the state vector; $S_a$ is the *a priori* covariance matrix for the state vector; $S_\varepsilon$ is the measurement error covariance matrix, which is assumed to be a diagonal matrix constructed using the spectra noise estimates. Noted that similar to the CO retrieval algorithm (**Zeng et al., 2022**), we have enlarged the spectra noise by a factor of 2.0 such that the averaged reduced $\chi^2$ value from the optimal estimation $NH_3$ retrieval is close to 1.0. This extra noise represents the unaccounted uncertainty from the forward model and absorption spectroscopy by the original instrument noise alone.

A "perfect" observing system, which has sufficiently good sensitivity to each element in the retrieval vector, would have an AK matrix close to an identity matrix by theory. In reality, the detectivity is limited by various factors including the spectral noise, a priori uncertainty, and the sensitivity of the spectra to the geophysical variables in the state vector. As a result, the AK can be very different from an identity matrix. In general, the information from the true state is smoothed vertically over different layers by the retrieval algorithm. In this case, the rows of AK represent the smoothing functions. As described in **Rodgers (2000)**, the trace of the AK matrix is defined as the DOFS, which represents the number of independent elements of information extracted from the spectra by the retrieval algorithm for constraining $NH_3$. DOFS is an important metric that quantify the vertical resolution of the retrieval profile. For example, a DOFS of 1.0 means that, given the assumed $S_a$, at least one independent piece of information can be retrieved from the spectral measurement to constrain the vertical distribution of $NH_3$. Note that the DOFS is highly dependent on the magnitude of the assume $S_a$, an indicator of the a priori knowledge. If $S_a$ characterizes a weaker constraint, indicating less a priori knowledge, the DOFS will be higher as relative more information will be taken from the measurement.

### 3.3 The a priori $NH_3$ profile and covariance matrix

A fixed a priori is preferred for the purpose of this study for two reasons: (1) a fixed a priori eases the interpretation of the results compared to a time varying a priori. Any changes seen in the spatial and temporal patterns in $NH_3$ relative to the a priori reflects the information gained from the FY-4B/GIIRS spectra; (2) it is not applicable to get a reasonable a priori estimate for all hours in a day from just the spectra (e.g., using channel brightness temperature difference as in **Shephard et al. (2011)** and **Warner et al. (2016)**) without relying on model simulations, because the diurnal change of TC that affects GEO satellite observation is much more complex than that in the two overpasses for LEO satellite each day. The single a priori $NH_3$ profile, as shown in **Figure 3(a)**, for all retrievals in the retrieval algorithm is derived from $NH_3$ simulations from the Goddard Earth Observing System composition forecast (GEOS-CF; **Keller et al., 2021**) model developed by NASA's Global Modeling and Assimilation Office (GMAO). One year of simulation in 2022 is used to get the mean and standard deviation of $NH_3$ vertical distribution. To avoid over sampling of the background regions, only simulations in the representative land regions in east Asia (20°-60°N and 110°-120°E) and south Asia (20°-40°N and 70°-100°E) are used. The negative value toward the lower end of the error bar does not have physical meaning, it is caused by the large standard deviation derived from model simulations that do not strictly follow a normal distribution. The a priori total $NH_3$ column is about $1.5 \times 10^{16}$ molecules/cm$^2$. To construct the correlation matrix, we used a correlation length of 3 km based on our analysis of the GEOS-CF reanalysis. Most of the layers show correlation lengths between 1 to 3 km and we use upper bound (3 km) to increase the stability of the retrieval system. The covariance matrix calculated based on the a priori error and the correlation matrix is shown in **Figure 3(b)**.

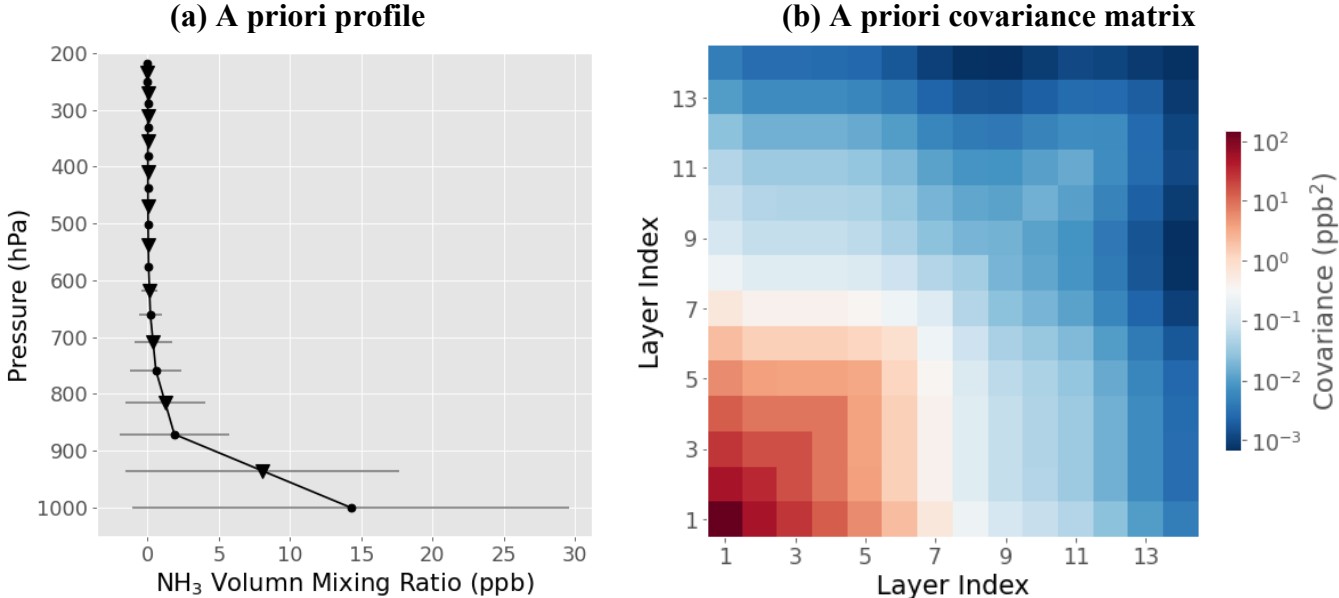

**Figure 3. (a) The a priori NH₃ profile used for retrievals of NH₃ in the FY-GeoAIR algorithm for FY-4B/GIIRS. This profile is computed from the GEOS-CF NH₃ simulations (Keller et al., 2021) in 2022 in representative land regions in east Asia (20°-60°N and 110°-120°E) and south Asia (20°-40°N and 70°-100°E). The error bars represent one standard deviation for different layers from all NH₃ profiles; (b) The a priori covariance matrix constructed from the error estimate in the a priori profile and the correlation matrix with a correlation length of 3 km. See text for the details.**

### 3.4 Post-filtering of NH₃ retrievals

After cloud screening, there are in total of 11.7 million clear-sky data points for the 6 months from July to December of 2022. In the post-processing, multiple filters are applied to ensure good retrieval quality. First, retrievals that fail to converge after 10 iterations are excluded. Second, retrievals with the goodness of fit, quantified by reduced $\chi^2$, less than 1.5 are excluded. Lastly, retrievals with RMSE of fitting residual larger than 0.25 K are excluded. After post-filtering, about 10.5 million data points pass the filters. The histograms of the reduced $\chi^2$ and spectral fitting residual are shown in **Supplementary Figure S1** (after filtering) and **Figure S2** (before filtering). The average reduced $\chi^2$ after post-filtering is 0.76, suggesting a satisfactory goodness of fit. In the following analysis, an extra filter based on DOFS may apply to exclude data with low DOFS and, therefore, low information content extracted from the observed spectra.

### 3.5 Retrieval experiments for quantifying retrieval error

A synthetic experiment is carried out by (1) generating simulated synthetic spectra based on pre-determined NH₃ profiles which can be regarded as the "truth", (2) adding assumed noise according to the spectra noise of FY-4B/GIIRS, and (3) applying the FY-GeoAIR algorithm to retrieve NH₃. By comparing the retrieval to the "truth", we can evaluate the performance of the retrieval algorithm and its relationship with relevant driving factors. In this experiment, the "truth" NH₃ profiles are extracted from the GEOS-CF model simulations on six representative days: July 07, August 05, September 06, October 10, November 15, and December 18 of 2022, when the available number of clear-sky observations is among the largest in the specific month. By randomly sampling 500 data points for each observation cycle from the FY-4B/GIIRS clear-sky observations, we carried out 36000 simulations in total. The retrieved NH₃ columns are finally compared with the "truth" and correlated with DOFS, as shown in

**Figure 4**. Simulations with DOFS less than 0.3 are not shown due to their high uncertainty and low information content extracted from the spectra measurement. The scatter plot suggests that the consistency between the retrieval and the "truth" increases when the DOFS becomes larger. For retrievals with low DOFS, the retrieval values are close to the a priori value, which is about $1.5 \times 10^{16}$ molecules/cm$^2$. For the several outliers that have high DOFS but with poor retrieval, it is likely because the "truth" NH$_3$ profile structure is far away from the a priori profile structure. For DOFS>0.5, the RMSE of the retrieval is about $1.67 \times 10^{16}$ molec/cm$^2$,

while for DOFS>0.7, the RMSE reduces to $1.37 \times 10^{16}$ molec/cm$^2$, representing error for a single retrieval. Since the retrieved NH$_3$ columns have a monthly mean between $2.0 \times 10^{16}$ and $6.0 \times 10^{16}$ molec/cm$^2$ (as shown in **Section 4.2)**, this synthetic experiment results indicate that the retrieval error is on average between 23% and 68% for a single retrieval when DOFS>0.7. On the other hand, when an ensemble mean (e.g., monthly mean) of NH$_3$ columns is derived, the resulting error for the mean can be much smaller (reduced by a factor of sqrt(N) where N is the number of observations) when a large number of observations is available.

In addition, two more experiments have been carried out to (1) compare the difference in retrievals from a different micro-window (920-940 cm$^{-1}$) to investigate the possible impact of spectral noise, and (2) investigate the impact of a priori NH$_3$ vertical shape on the retrievals. The details are described in the **Supplementary Text S1** and **Text S2**, respectively. The results show that the DOFSs show high consistencies, with a correlation coefficient of 0.97, suggesting the two micro-windows contain similar information in capturing the NH$_3$ variabilities. The correlation coefficient between the two retrieved NH$_3$ column datasets is 0.81

with a root-mean-square-error of $3.2 \times 10^{15}$ molec/cm$^2$, suggesting the spectral noise is not causing large bias. The results show that the mean and standard deviation of the fractional error are 1.0% and 9.65%, respectively, for the reduced PBL excess profile, and 0.9% and 7.6%, respectively, for the enhanced PBL excess profile. Fortunately, there are no large systematic bias and the averaged error is within 10% in our cases of profiles differ by a factor of 2.

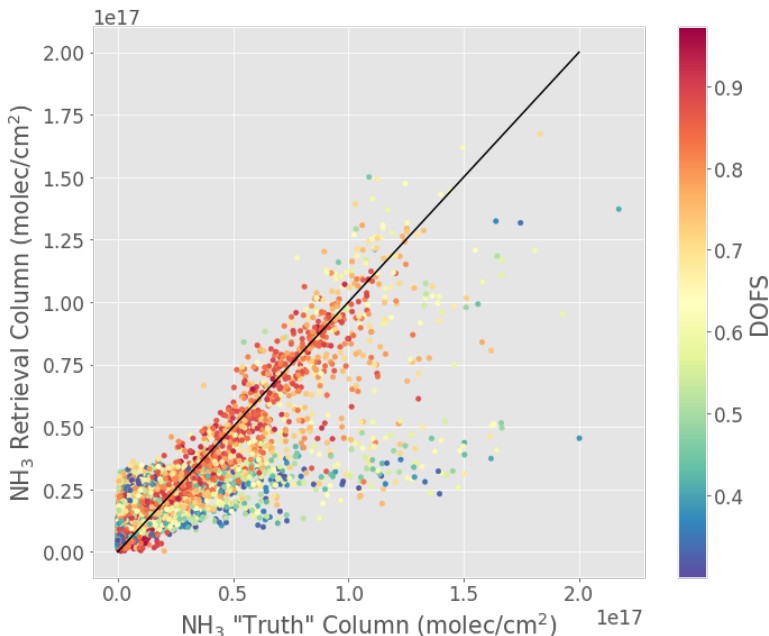

Figure 4. Comparison of retrieval and the "truth" NH$_3$ columns from the synthetic experiment. The retrieved NH$_3$ columns

are based on simulated spectra generated based on pre-determined NH$_3$ profiles ("truth"). The data points have been filtered by DOFS>0.3 and are color-coded by the corresponding DOFS value. The RMSE of the retrieved NH$_3$ columns relative to the "truth" is about $1.67 \times 10^{16}$ molec/cm$^2$ for DOFS>0.5 and $1.37 \times 10^{16}$ molec/cm$^2$ for DOFS>0.7.

## 4. Results and discussion

### 4.1 Information content analysis based on DOFS and AK matrix

This section conducted information content analysis by investigating the spatial and diurnal changes of DOFS from NH$_3$ retrievals, and how the DOFS changes are related to TC. In addition, we examine the diurnal changes of the vertical sensitivity as quantified by AK matrix. In particular, we focus on three representative regions including source regions in North China Plain, North India and a background region in Mongolia.

The spatial maps of DOFS, as shown in **Figure 5**, for different times in a day clearly show the spatial gradient and diurnal change. In July and October, the DOFSs in the afternoon is usually the highest and those in the evening the lowest. In December, night-time DOFS are significantly higher than other seasons. Previous studies by **Clarisse et al. (2010, 2021)** and **Bauduin et al. (2017)** using IASI observations have shown that the DOFS is primarily driven by the TC. When thermal contrast is close to zero, measurement sensitivity is low, and DOFS are close to zero. Large positive TC increases sensitivity and results in NH$_3$ spectral signatures that are seen in absorption. Large negative TC also allows for sensitive measurements, this time allowing NH$_3$ spectral signatures to be seen in emission. Negative TC corresponds to the situation where the atmosphere is warmer than the surface, allowing to decorrelate the surface layer with the rest of the lower troposphere. In addition, higher concentration of target gases provides stronger detectivity than lower concentration. The relationships between TC and DOFS are illustrated in **Figure 6** for the three representative regions, including (1) North China Plain, which represents industrialized and agricultural regions with persistently high NH$_3$ emissions; (2) North India, which represents another important NH$_3$ source region in Asia; and (3) Mongolia, which represents a NH$_3$ background region. A typical "butterfly" shape can be seen in almost all cases, except for North India in July when there are much less observations due to clouds. In general, the DOFS increases when the TC becomes either more positive or more negative, consistent with results from **Clarisse et al. (2010)** and **Bauduin et al. (2017)** based on polar-orbiting satellite. In NH$_3$ source regions (e.g., North China Plain and North India), the DOFSs are higher for the same TC compared with non-source region (e.g., Mongolia), suggesting the contribution from higher NH$_3$ concentration to the total information content.

This strong correlation between DOFS and TC or NH$_3$ abundance is also reflected in the spatial maps of DOFSs in **Figure 5** when analyzed with the corresponding TC maps in the **Supplementary Figure S2**. The source regions in North China Plain and North India have higher DOFSs than other non-source regions, such as Tibetan Plateau although it has large TC. For the same region (e.g., North China Plain), the higher DOFS is driven by the more positive TC in the afternoon in July, while in the nighttime the TC is closer to zero that leads to much lower DOFS. As it approaches the winter season, from October to December, the diurnal cycle of TC, which shifts from positive in the daytime to negative in the nighttime, gradually become stronger. The changes are basically driven by the faster warming or cooling properties of the land compared to the atmosphere. Fortunately, both situations favor the detection of NH$_3$ using thermal infrared and lead to the high DOFSs in both the daytime and nighttime in both October and December.

As mentioned above, measurement sensitivity of NH$_3$ is driven by TC and the NH$_3$ abundance (**Clarisse et al., 2010**). This is also illustrated in the **Appendix Figure A1(a)** and **(b)** for a large positive and negative TC. As described above, while a positive TC leads to stronger absorption features, a negative TC causes spectral emission features allowing the detection of NH$_3$ also during the night (see also the example GIIRS spectra shown in **Clarisse et al. (2021)**). In both cases we see that the averaging kernels peak at the surface and the posteriori uncertainty in the retrievals of the surface layer are largely reduced compared to the a priori uncertainties. However, when the TC is small, as in the **Appendix Figure A1(c)**, the DOFS values become smaller, and the AVK peaks higher up in the atmosphere (in this case, in the second layer). The retrieved value remains close to the a priori and the posteriori error is almost the same as the a priori, indicating low information content of the measurement. These examples illustrate the importance of TC for infrared sounding of boundary layer NH$_3$. An important advantage of GEO compared to LEO IR sounders

is that they make observations throughout the day, such that optimal measurement conditions (large TC) can be found more readily. The diel variations of TC and DOFS are illustrated in the **Appendix Figure A2** for North China Plain and North India. LEO IR sounders like IASI with an equator crossing times at 9:30 am and 9:30 pm LT in general do not measure at the time where measurement sensitivity (or DOFS) is largest. The optimal time is found around noon.

The short lifetime of $NH_3$ of hours to days means that the $NH_3$ concentration is highly concentrated in the planetary boundary layer (PBL) around the source region. This is also illustrated by the significant higher concentration below 800 hPa in the a priori $NH_3$ profiles (**Figure 3**). As a result, observation of $NH_3$ from thermal infrared has high sensitivity closer to the surface compared to relatively longer-lived air pollutant such as carbon monoxide. The vertical sensitivities of FY-GeoAIR $NH_3$ retrievals are shown in **Figure 7** for the three representative regions using the corresponding averaging kernel diagonal vectors, which are measures of the DOFS for each vertical layer. We can see the diurnal AK values peak at the surface layer for North China Plain for all months, especially for October and December, due to the high $NH_3$ concentration and favorable TC changes. In July, the nighttime sensitivity is significantly reduced in the surface layers. In North India, the bottom layers sensitivity gets to the highest at midnight of October and December due to highly negative TC. In July, the TC closing to zeros leads to much lower AK values in North India. In Mongolia, the low background $NH_3$ concentration means the AK values are low over all vertical layers. The small changes in DOFS are primarily driven by the diurnal change of TC in this case. Overall, the $NH_3$ retrievals using FY-GeoAIR algorithm from FY-4B/GIIRS observations show high detectivity in the surface layer, especially for source regions in North China Plain and North India.

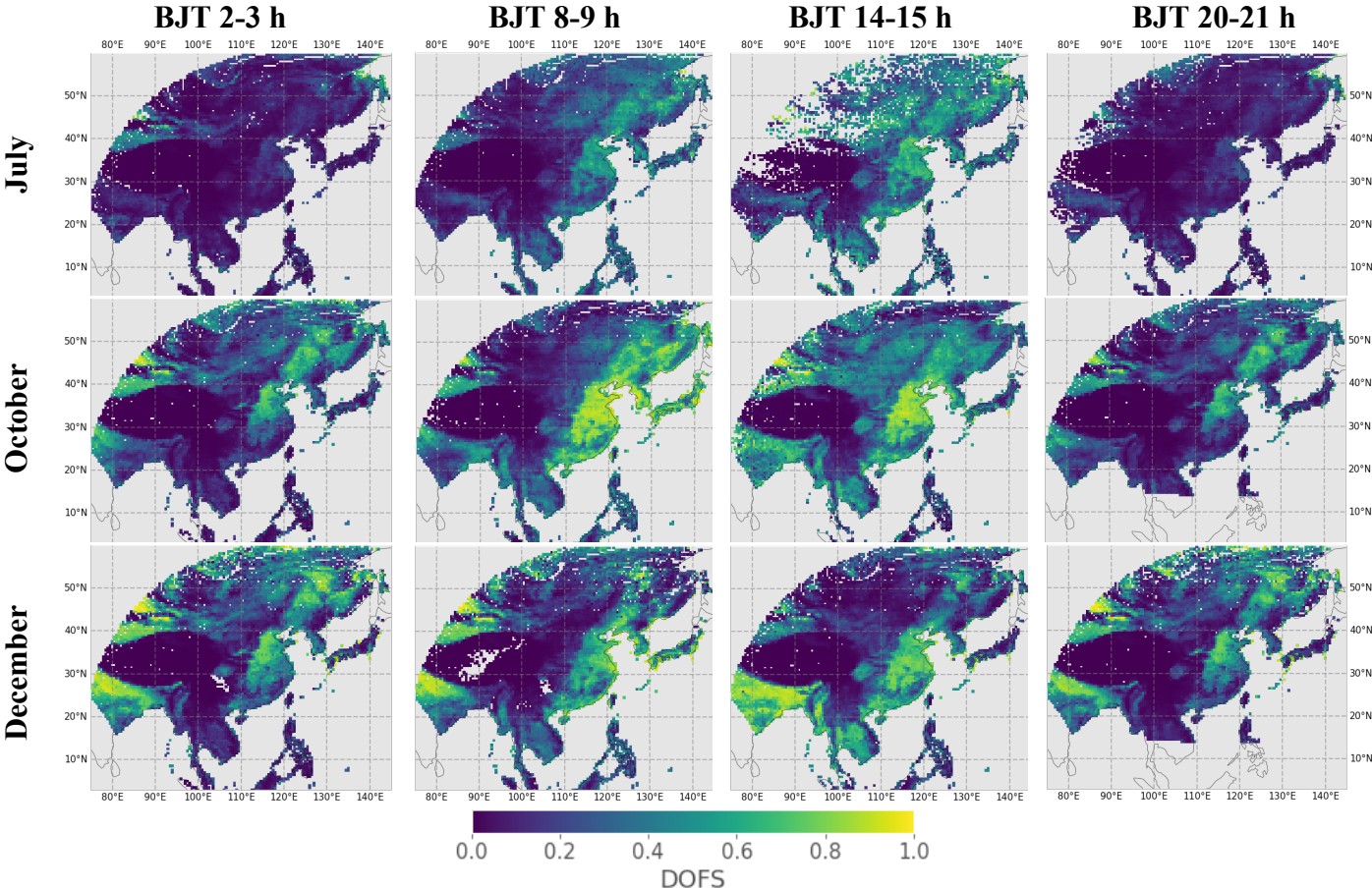

**Figure 5. Distributions of DOFS averaged for July (summer), October (autumn), and December (winter), respectively, in 2-3h, 8-9h, 14-15h, and 20-21h in Beijing Time (BJT) to represent mid-night, early morning, afternoon, and early evening, respectively. The corresponding maps for TC are presented in Supplementary Figure S3.**

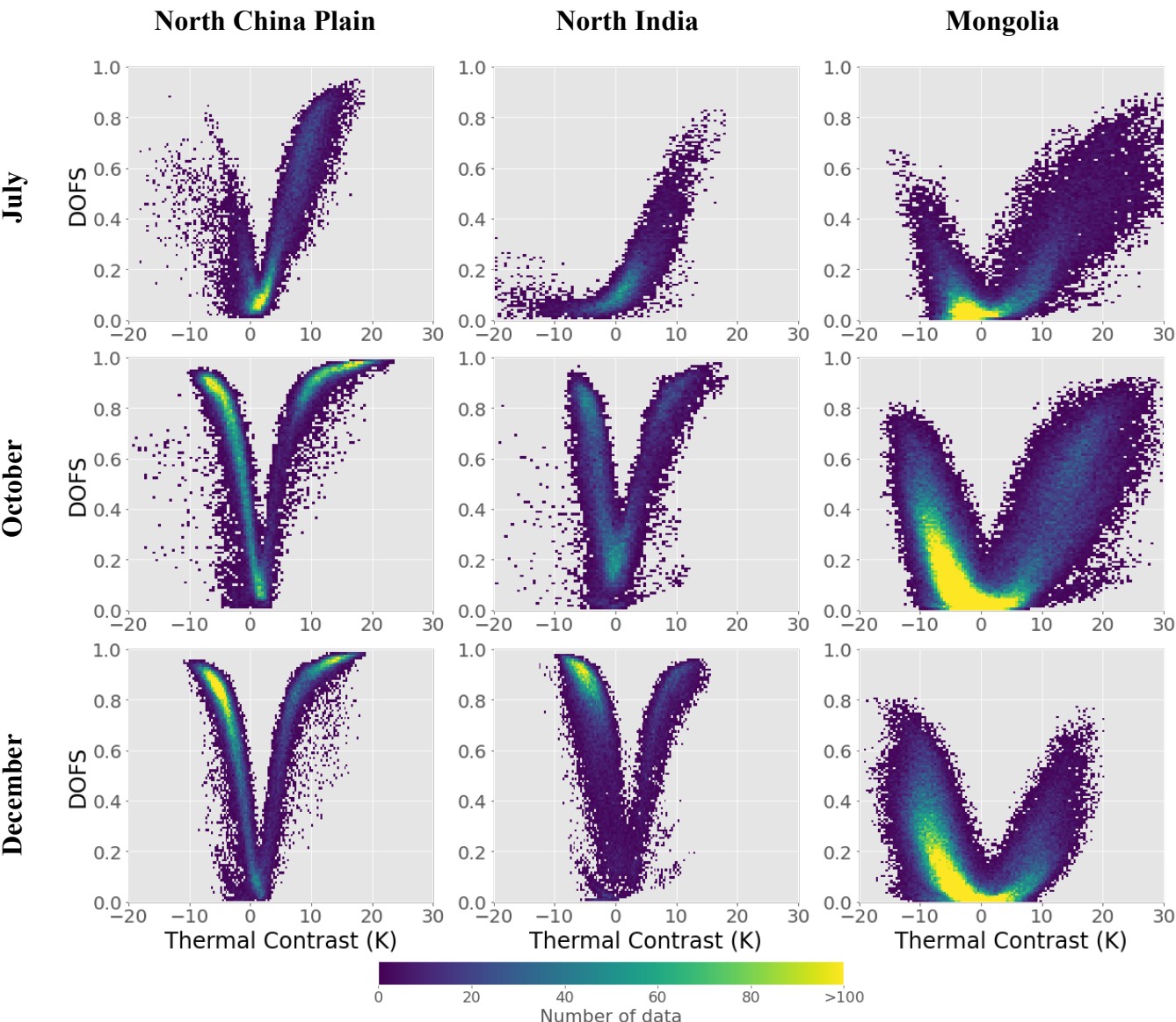

Figure 6. Scatter plots between TC and DOFS from NH$_3$ retrievals in North China Plain, North India, and Mongolia, respectively, for July (summer), October (autumn), and December (winter). The coverages of the three representative regions are 32°-40°N and 115°-120°E for North China Plain, 22°-27°N and 77°-87°E for North India, and 40°-50°N and 100°-110°E for Mongolia.

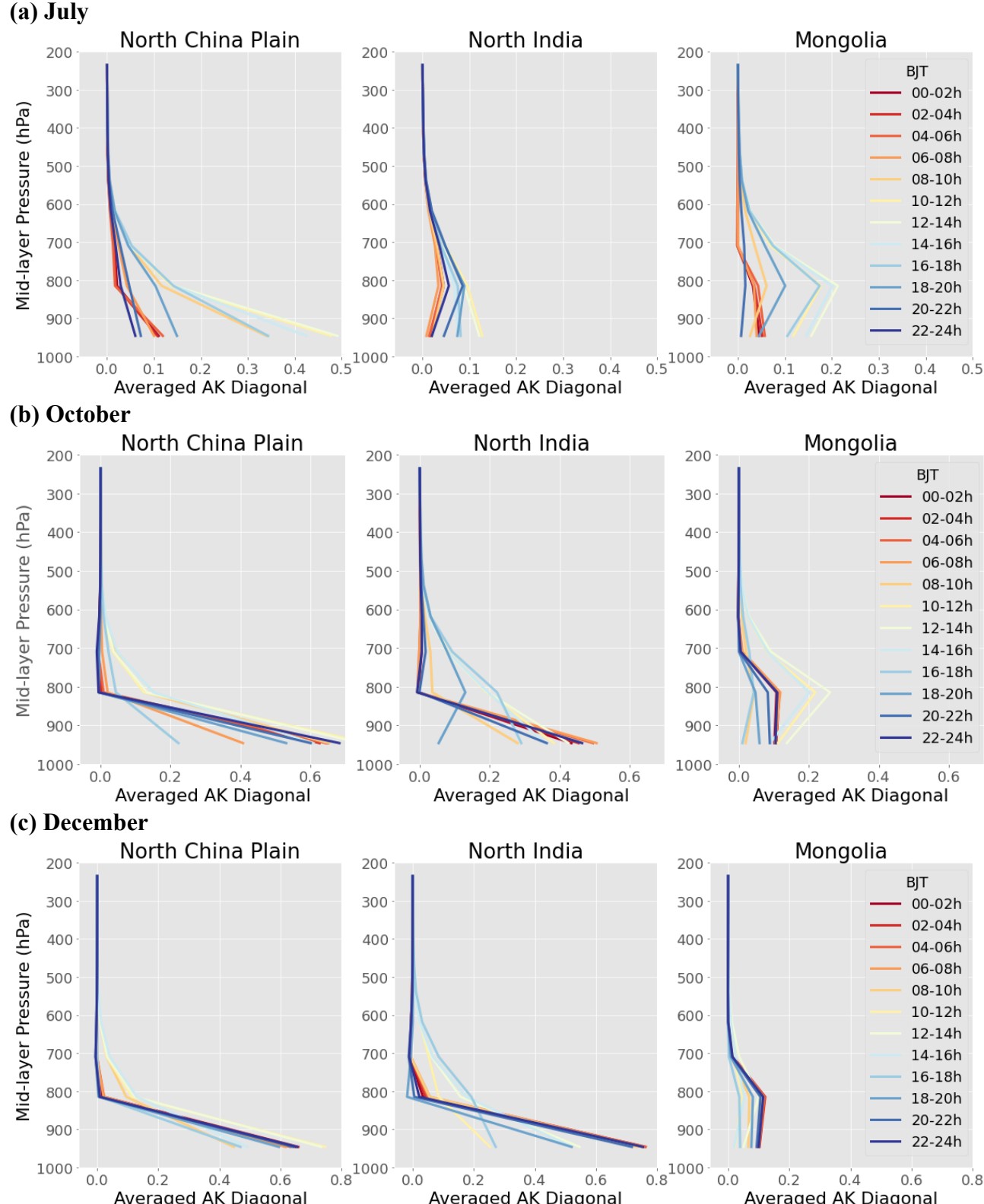

**Figure 7. Averaged averaging kernel diagonal vectors for the three representative regions in North China Plain, North India, and Mongolia for July (summer, a), October (autumn, b), and December (winter, c). The averaging kernel rows are averaged for the 2-hour duration in each measurement cycle. The coverages of the three representative regions are the same as in Figure 6.**

**4.2 Spatial distribution and diurnal change of NH₃ column from FY-4B/GIIRS retrievals**

Spatial distribution maps of NH₃ columns are averaged for every 4-hour periods in a day and re-gridded into 0.5 by 0.5 degrees for every two months: July-August, September-October, and November-December. Before the aggregation, the post-filtered NH₃ retrievals are further screened by the criteria of DOFS>0.5. The screened NH₃ retrievals from the two micro-windows are further averaged for the mapping. The results are shown in **Figures 8** and **9**. The data gaps, especially in the Tibet Plateau, result from data filtering due to the low DOFS. The cloud filtering has considerably impacted data availability, in particular in North India in July-August during the Monsoon season. The nighttime data in summer have been mostly filtered due to their low TC and as a result low DOFS. From these maps, obvious diurnal cycle of NH₃ columns can be seen from all months. North China Plain and North India, which are two agricultural intensive regions with irrigated crops and a high density of livestock, show the highest values over the region. As explained in **Wang et al. (2020)**, the causes of high NH₃ loading in North India are due to the weak chemical loss and weak horizontal diffusion in North India, which are slightly different from those in the North China Plain.

The general diurnal cycle of NH₃ columns can be primarily explained by three possible driving factors, as concluded in the summary in **Clarisse et al. (2021)**, including the day-night difference in agriculture activities as a major source of NH₃, the temperature dependence of NH₃ emissions driven by diurnal and seasonal temperature changes, and the conversion between NH₃ gas and particulate driven by the day-night change of meteorological conditions. These can be used to interpret the quantitative analysis of the diurnal cycle as shown in **Figure 10** for North China Plain and North India. The corresponding diurnal changes of TC and DOFS are shown in the **Appendix Figure A2**. The day-night contrast of NH₃ columns from IASI onboard MetopB are also indicated to check if the day-night gradients between IASI and GIIRS are consistent. Specifically, the more significant diurnal cycle in summer (July-August) in North China Plain, captured by both GIIRS retrievals, can be explained by the higher temperature related emissions from plants and soils, and stronger daytime emissions from agricultural activities (**Meng et al., 2008**). In addition, the relatively low temperature and higher humidity in the nighttime, relative to the daytime, contribute to the conversion from NH₃ to particulates that leads to a lower NH₃ concentration. In North India, unfortunately, not sufficient data are availabel in July-August after post-filtering. In September-October and November-December, the diurnal cycles become less significant compared to the summer months as the main driving factors become less important. Interestingly, the winter diurnal cycle in North China Plain and North India show opposite pattern to the diurnal cycle in summer. The slightly increase of NH₃ columns in the nighttime, captured by both GIIRS and IASI retrievals, may be due to the shallower nocturnal boundary layer that traps the surface emissions (**Tevlin et al., 2017; Clarisse et al. 2021**). To ensure the diurnal change is not affected greatly by data quality, we also compare the GIIRS results using data with DOFS larger than 0.5 and 0.7, respectively, two thresholds that are high enough to ensure the quality of the retrievals. We can see that the difference in the diurnal cycle of NH₃ columns from FY-4B/GIIRS retrievals is not significant and therefore different data filters do not affect the general patterns of NH₃ diurnal cycle. Note that the variability of NH₃ columns within the region is large as shown by the error bars (one standard deviation). This large variability is a result of NH₃'s short life time and the spatial heterogeneity of its emissions. Moreover, when compared with the averaged uncertainty of a single retrieval ($1.37 \times 10^{16}$ to $1.67 \times 10^{16}$ molec/cm$^2$ as derived from the retrieval experiment in **Section 3.5**), the day-night contrast of the averaged diurnal variations of NH₃ columns as shown in **Figure 10** may not be significant for the North China Plain in September-October and the North India in November-December.

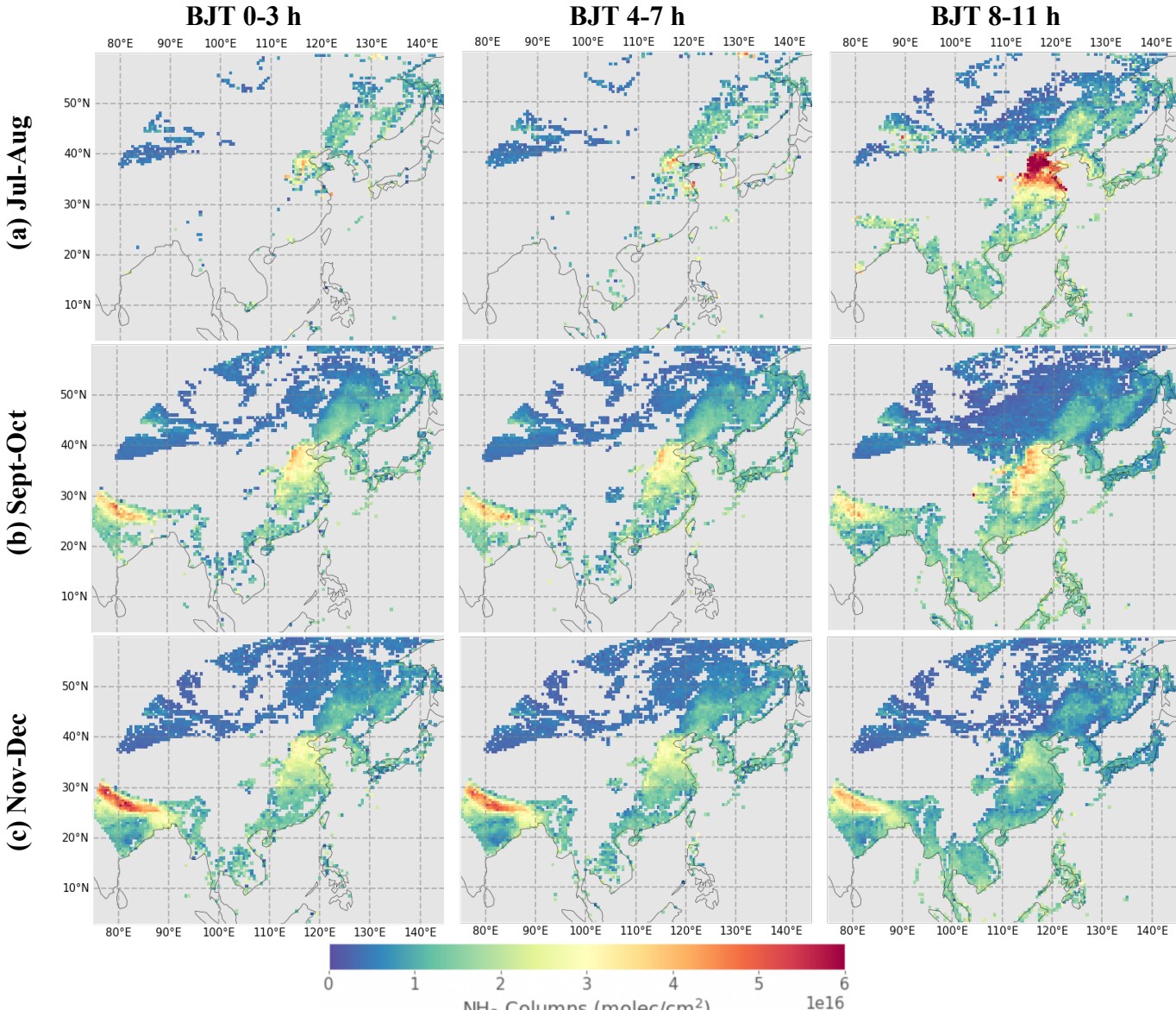

**Figure 8. Monthly maps of NH₃ columns averaged for every four-hour (0-3 h, 4-7 h, and 8-11 h in Beijing Time) and two-month for July-August, September-October, and November-December of 2022. These NH₃ retrievals are further filtered by DOFS>0.5 and then re-gridded into 0.5 by 0.5 degrees.**

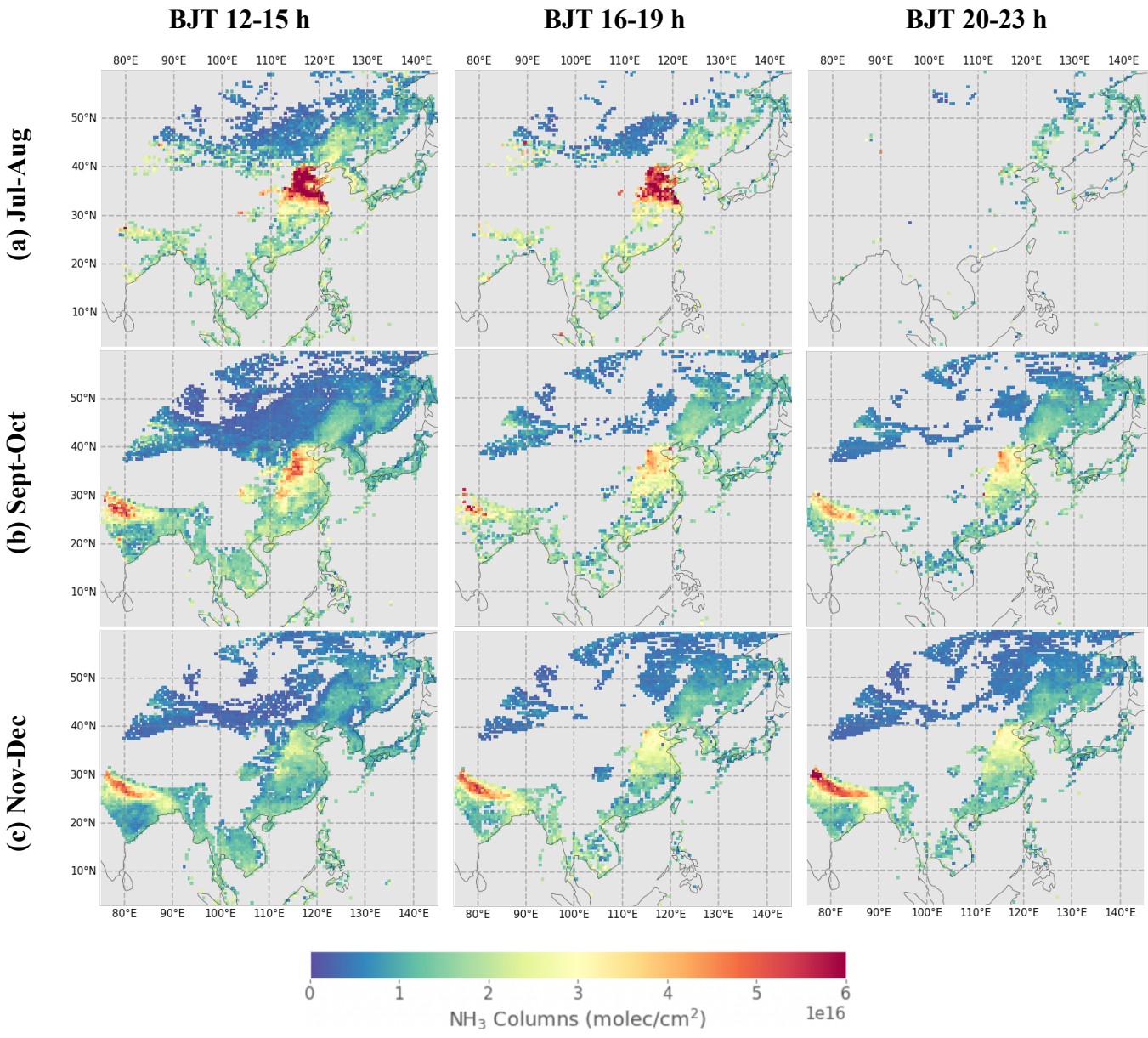

**Figure 9. Same as Figure 8 but for 12-15 h, 16-19 h, and 20-23 h in Beijing Time.**

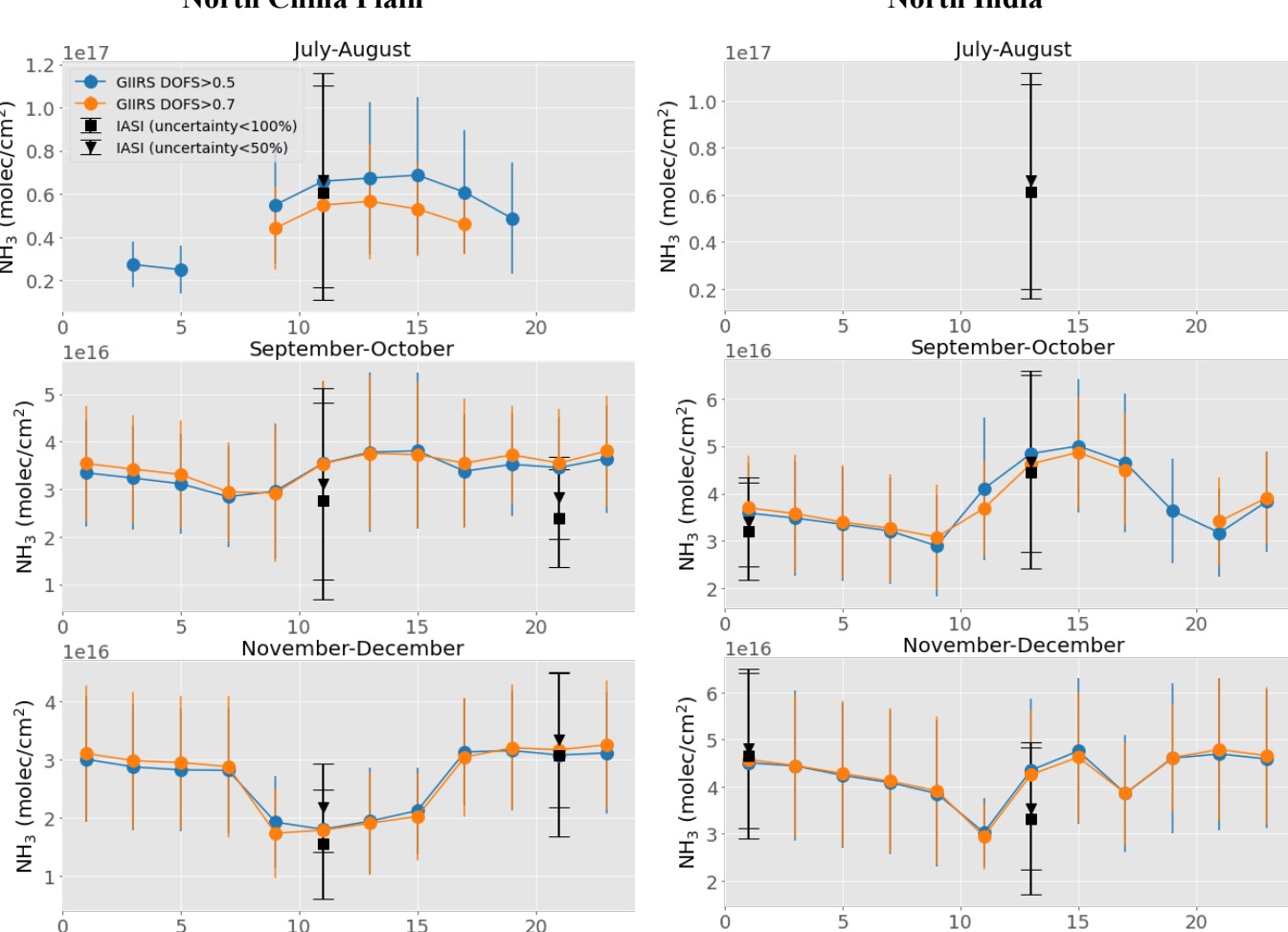

**Figure 10. Diurnal cycle of NH₃ columns from FY-4B/GIIRS retrievals. The NH₃ columns are from retrievals using the micro-windows of 955-975 cm⁻¹ for the 2-hour duration in each measurement cycle when at least 30 data points are available. The error bar represents one standard deviation. Diurnal cycles based on two different filtering criteria (DOFS>0.5 and DOFS>0.7) are shown simultaneously. The IASI NH₃ column retrievals (two over-passing times in a day) are averaged values in the corresponding months. IASI NH₃ retrievals with uncertainty larger than 50% are not used. Data are averaged in two-month periods, when at least 10 data points are available. Note that the coverages of North China Plain (36-38°N; 116-118°E) and North India (26-28°N; 78-80°E) are smaller than Figure 6 to focus on the central source regions.**

**4.3 Comparison with IASI NH₃ retrievals**

IASI onboard Metop-B measures $NH_3$ two times a day at 9:30 am and 9:30 pm LT (equator crossing times), respectively, in the morning and evening. The $NH_3$ data from IASI are retrieved based on hyperspectral radiance index (HRI) and a trained neural network that relates HRI to $NH_3$ distributions (**Whitburn et al., 2016**). Subsequent improvements to the algorithm and data product are outlined in **Van Damme et al. (2017) and Van Damme et al. (2021)**. For the comparison, the latest version 4 (**Clarisse et al., 2023**) of the IASI is used, with data originating from IASI/Metop-B from July to December, 2022. We filtered out the data with cloud cover larger than 20%. Note that the a priori vertical profile of $NH_3$ in IASI is different from the a priori profiles for the FY-4B/GIIRS retrievals. The a priori in IASI algorithm assumes a Gaussian shape with peak altitude set at the surface over land. The width of the Gaussian shape equals to the boundary layer height. Two different comparisons are carried out, as follows.

**(1) Comparison of spatial distribution on representative days**

We first take a look at the spatial distribution of $NH_3$ columns on two representative days: July 07, 2022 (summer, local daytime) when there was mostly positive TC, and December 18, 2022 (winter, local nighttime) when there was mostly negative TC. The retrievals are resampled on a 0.5°×0.5° grid. The daily mean value for each grid cell is computed when at least 3 points are available. Only GIIRS retrievals with DOFS>0.2 and IASI retrievals with a relative uncertainty below 50% are used. The results are shown in **Figure 11**. A good agreement is found in general with fitted slopes close to unity, with a larger scatter for the summer case. The IASI $NH_3$ tends to be higher than the GIIRS retrievals, especially for high values.

**(2) Collocated point-by-point comparison**

We further conduct a spatially and temporally collocated point-by-point comparison between FY-4B/GIIRS and IASI $NH_3$ retrievals. The results are shown in **Figure 12** and the **Appendix Figure A3**. Again, only GIIRS retrievals with DOFS>0.2 and IASI retrievals with a relative uncertainty below 50% are used. We consider observations to be collocated when the distance between the centers of the pixels is less than 6km (half of the FY-4B/GIIRS footprint size) and the observation time difference less than 1 hour. All data points that meet these criteria are used for the comparison. The correlation coefficient (r), root-mean-squre-error (rmse) and the fitting slope are also indicated. We can see the comparison shows good agreement with high correlation coefficient. The retrievals are highly consistent especially for winter months. The nighttime data in July has mostly been screened due to low DOFS. Except for the daytime data in summer (July and August), both retrievals agree within the estimated error as quantified by RMSE in **Section 3.5**. The systematic bias in the daytime in July and August is probably due to the different a priori of $NH_3$ used in the algorithms. Since the IASI a priori profile changes with boundary layer height, the a priori profiles have different vertical structure depending on the time of the year, compared to the GIIRS a priori which uses a fixed average across the year. To test this hypothesis, we carry out an experiment to generate a new set of GIIRS $NH_3$ retrievals in July daytime that use the Gaussian shape a priori profiles following IASI algorithm (**Clarisse et al., 2023**). We keep the total column for the new a priori profile unchanged. The results are shown in the **Appendix Figure A4**. It shows that the agreement in $NH_3$ total column between IASI and GIIRS has largely improved, especially for the retrievals that have DOFS>0.5. The bias has significantly reduced. This result demonstrates that the systematic bias between GIIRS and IASI is caused by the difference in their a priori profile structure.

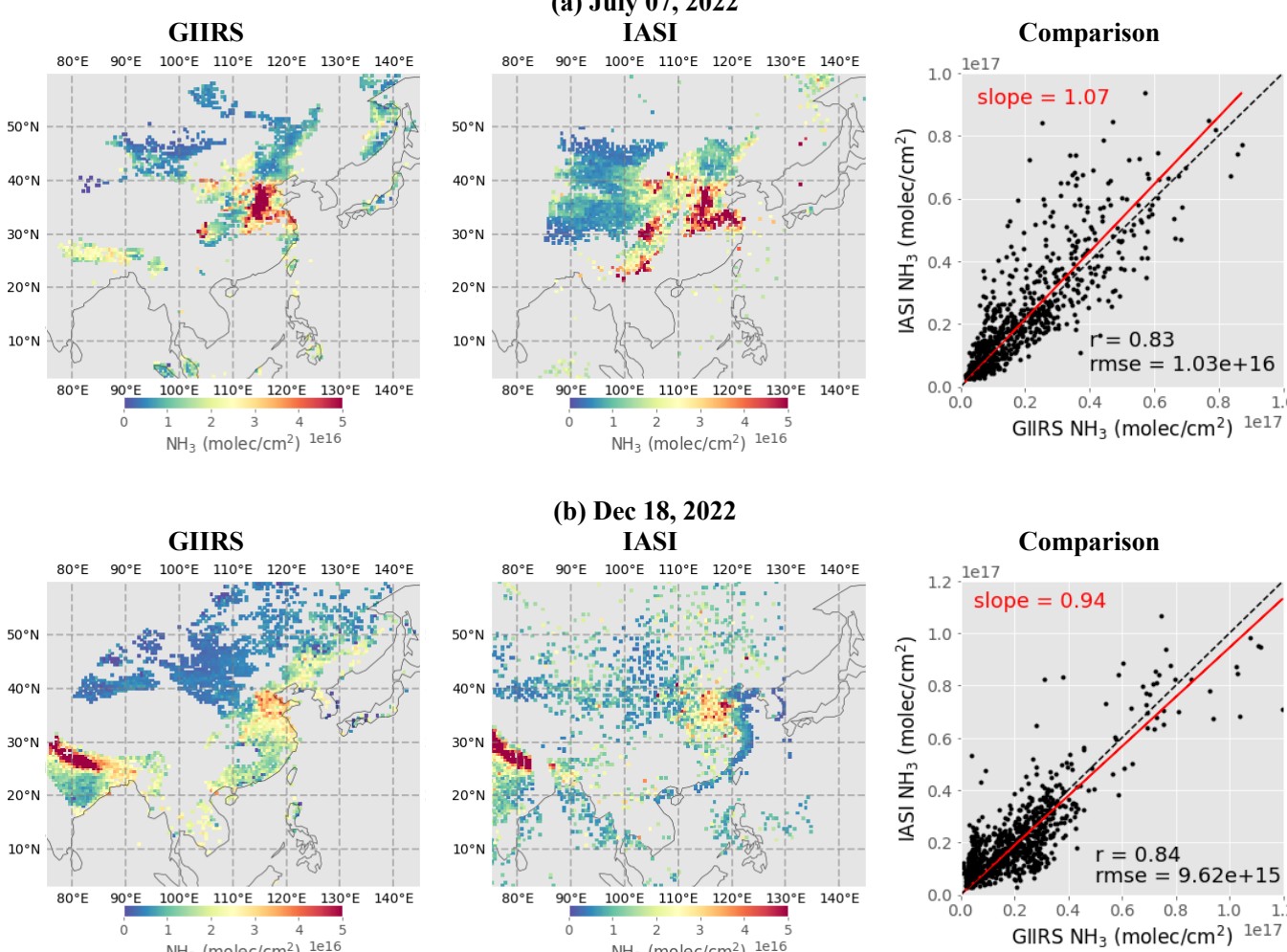

**Figure 11. Gridded intercomparison between NH₃ column retrievals from GIIRS (1st column) and IASI (2nd column) on two representative observations, July 07, 2022 daytime (upper panel, a) and December 18, 2022 nighttime (lower panel, b) retrievals. The observation hours of GIIRS are selected to be close to that of IASI, the observation hours are shown in the Supplementary Figure S7. The scatter plots (3rd column) show the comparison of collocated GIIRS and IASI averaged column data in each grid. The retrievals are re-grided to 0.5°×0.5° grids in the region. The daily mean value for each region is computed when at least 3 grid points are available.**

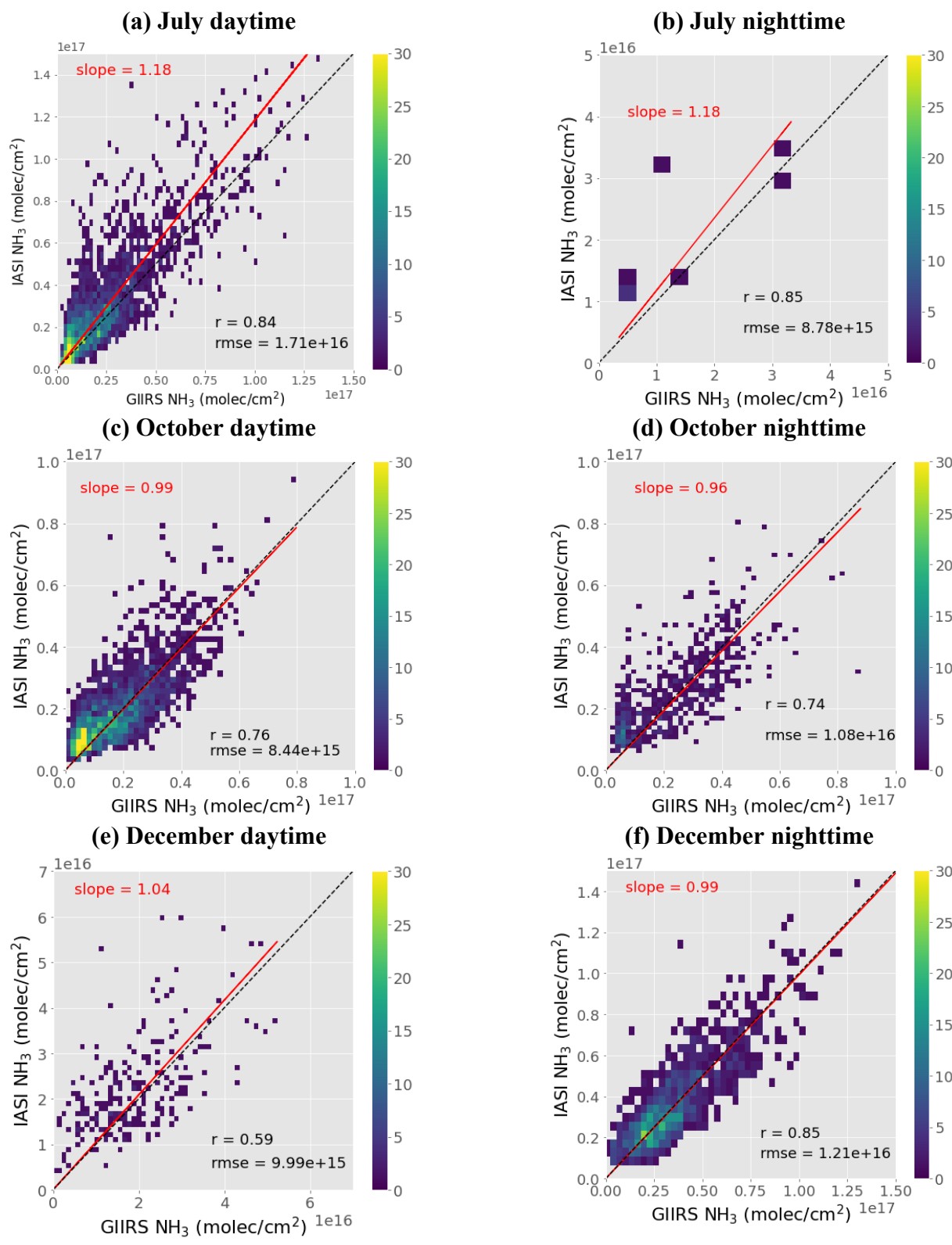

**Figure 12. Intercomparison of collocated GIIRS and IASI NH₃ columns for July, October, and December. Observations are considered to be collocated when observed within 6 km and time difference smaller than 1 hour. The correlation coefficient (r), root-mean-squre-error (rmse) and the fitting slope are also indicated. Similar results for August, September, and November are shown in the Appendix.**

## 5. Conclusions

In this study, we present an $NH_3$ retrieval algorithm based on the optimal estimation method for FY-4B/GIIRS. The DOFS and AK matrix produced from the retrieval algorithm are examined to evaluate the information content and vertical sensitivities in constraining the diurnal cycle of $NH_3$ in East Asia. Our retrievals are carried out using the $NH_3$ micro-window (955-975 cm$^{-1}$).
Retrieval results using FY-4B/GIIRS spectra from July to December 2022 show that (1) the detection sensitivity, as quantified by the AK matrix, peaks in the lowest 2 km atmospheric layers, which facilitates the observation of emission sources in the PBL; (2) the DOFS and TC are highly correlated, resulting in a typical "butterfly" shape, showing that the DOFS increases when the TC becomes either more positive or more negative; (3) the diurnal cycle of $NH_3$ columns from FY-4B/GIIRS show significant diurnal cycle in summer (July-August) in North China Plain, in good agreement with the day-night gradient from the collocated IASI
retrievals. The weak and moderate diurnal cycles in two important source regions of North China Plain and North India in September-October and November-December are also presented from both FY-4B/GIIRS and IASI retrievals. This study demonstrates the capability of GIIRS in observing the diurnal $NH_3$ changes in East Asia, making it a unique dataset for quantifying $NH_3$ emissions and depositions and evaluating strategies for managing anthropogenic sources of $NH_3$ in Asia.

As the world's first geostationary infrared sounder, GIIRS instruments onboard FY-4A and FY-4B provide unique hyperspectral
thermal infrared observations to quantify the diurnal change of atmospheric composition in East Asia. Other existing and in planning geostationary missions include South Korea's Geostationary Environment Monitoring Spectrometer (GEMS) launched in Feb. 2020 to measure air quality in Asia using ultraviolet and visible bands (**Kim et al., 2020**), ESA's Sentinel-4 mission onboard the Meteosat Third Generation Sounder platform that consists of the thermal InfaRed Sounder (IRS) that will measure profiles of temperature, humidity, and atmospheric composition, and the Ultraviolet Visible Near-infrared (UVN) spectrometer that will
monitor air quality trace gases and aerosols in Europe (**Ingmann et al., 2012; Holmlund et al., 2021**), and NASA's Tropospheric Emissions: Monitoring of Pollution (TEMPO) that will observe air quality in North America (**Zoogman et al., 2017**). These GEO missions form a global network that enables diurnal observation to cover global important emission sources, which will significantly enhance local and global air quality and climate research.

**Data availability**

The sample NH$_3$ retrieval data from FY-4B/GIIRS in this study are publicly available from the Peking University Open Research Data Platform at https://doi.org/10.18170/DVN/VJ4MLO; The full retrieval dataset (total data size larger than 150G) is available from the corresponding author upon request. FY-4B/GIIRS Level 1 data are publicly available from the FengYun Satellite Data Center at http://satellite.nsmc.org.cn/portalsite/default.aspx; The surface emissivity datasets are downloaded from the Global Infrared Land Surface Emissivity: UW-Madison Baseline Fit Emissivity Database at https://cimss.ssec.wisc.edu/iremis/; The ECMWF ERA5 reanalysis datasets are available from the Copernicus Climate Data Store at https://cds.climate.copernicus.eu/; The ECMWF atmospheric composition datasets are available from the Copernicus Atmosphere Data Store at https://ads.atmosphere.copernicus.eu/. NH$_3$ simulation data from GEOS-CF is downloaded from https://gmao.gsfc.nasa.gov/weather_prediction/GEOS-CF/data_access/. IASI is a joint mission of EUMETSAT and the Centre National d'Etudes Spatiales (CNES, France). The IASI NH$_3$ product is available from https://iasi.aeris-data.fr/nh3/.

**Acknowledgment**

Z.-C. Zeng acknowledges funding from the National Natural Science Foundation of China (grant no. 42275142 and no. 12292981), the National Key R&D Program of China (grant no. 2022YFA1003801), and the Fundamental Research Funds for the Central Universities at Peking University (grant no. 7101302981). This work was also supported by High-performance Computing Platform of Peking University. Research at the National Satellite Meteorological Center (NSMC) was funded by NSMC of China Meteorological Administration (CMA) under the program of Calibration Technology Development and Level-1 Data Production for the Hyperspectral Imaging and Sounding Instruments onboard FY-3E and FY-4B Satellites (FY-APP-2021.0507). Research in Belgium was co-funded by the Belgian State Federal Office for Scientific, Technical and Cultural Affairs (Prodex HIRS), the Air Liquide Foundation (TAPIR project), and ESA (Short-lived greenhouse gases CCI project). This work is also partly supported by the FED-tWIN project ARENBERG ("Assessing the Reactive Nitrogen Budget and Emissions at Regional and Global Scales") funded via the Belgian Science Policy Office (BELSPO). L.C. is Research Associate supported by the Belgian F.R.S.-FNRS.

**Author contribution**

Z.Z. designed the study, developed the forward model and retrieval codes, carried out the experiments and results analysis, and prepared the manuscript. L.L. and C.Q. provided guidance on using the FY-4B/GIIRS L1 spectra data and carried out experiments related to spectra uncertainty analysis. L. C. and M. V. D. provided IASI NH3 v4 data and guidance on comparing GIIRS and IASI data. All authors reviewed and proofread the manuscript.

**Competing interest**

The authors declare that they have no conflict of interest.

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

## Appendix A

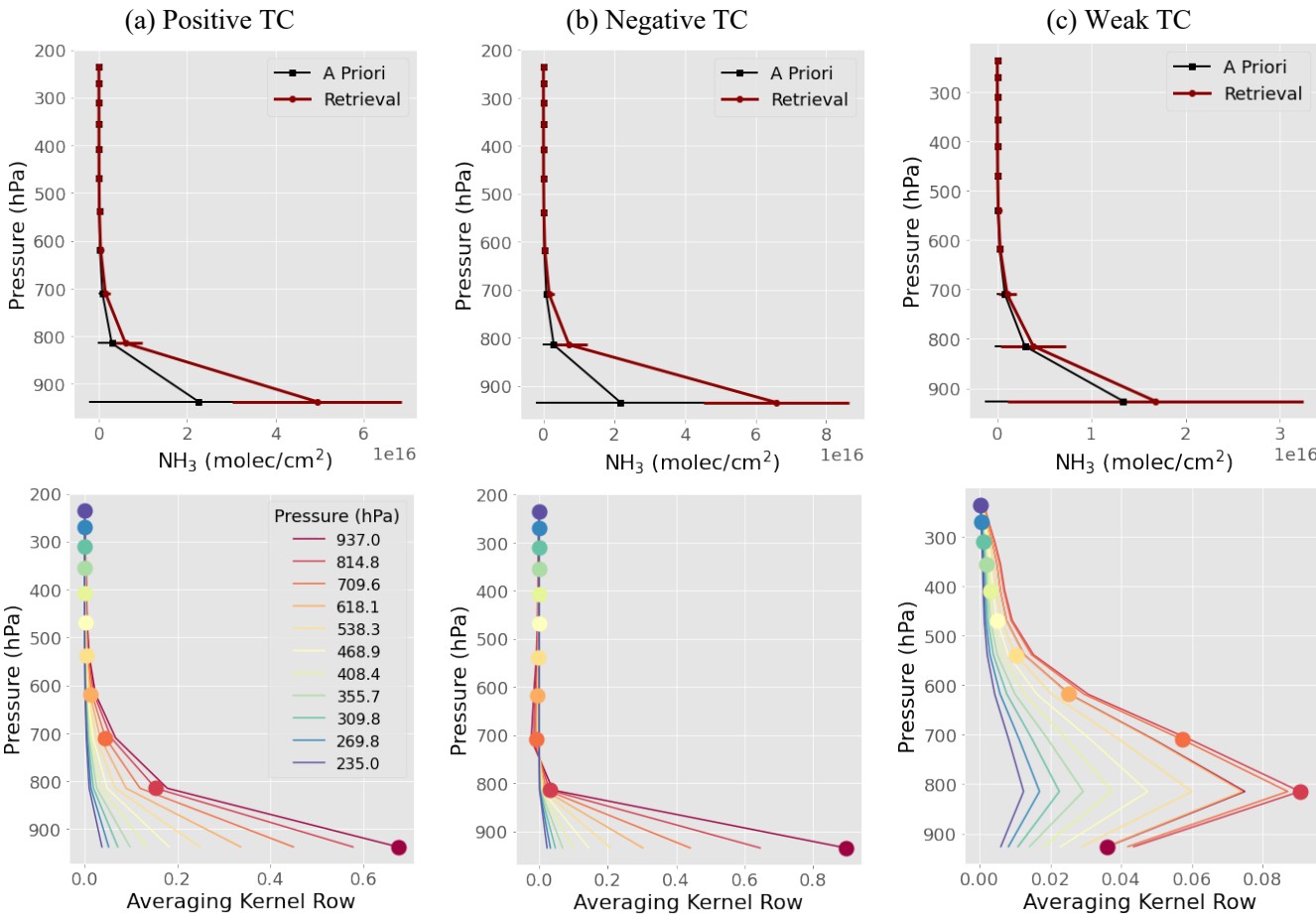

**Figure A1. Examples of the a priori and retrieval NH₃ profiles (upper panel) and the corresponding AK row vectors (second panel) for (a) positive TC with TC=13.37K and DOFS=0.89 from daytime measurement on July 07, 2022; (b) negative TC with TC=-6.7K and DOFS=0.91 from nighttime measurement on December 18, 2022; and (c) weak TC with TC=3.39K and DOFS=0.23 from early evening measurement on July 06, 2022.**


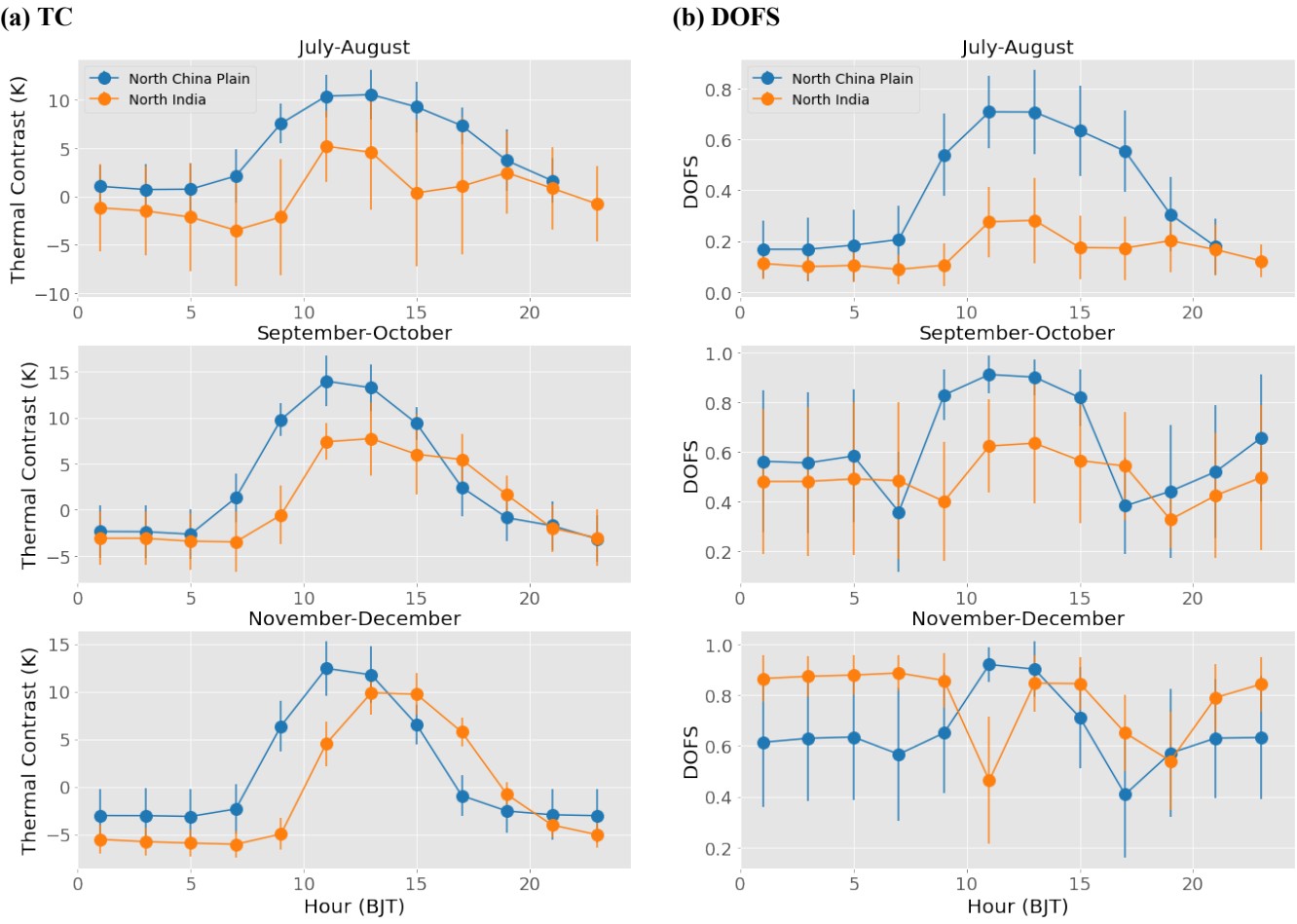

**Figure A2. Associated figure for Figure 10 on the diurnal change of TC and DOFS for the North China Plain and North India. Different from Figure 10, no extra filters have been applied.**

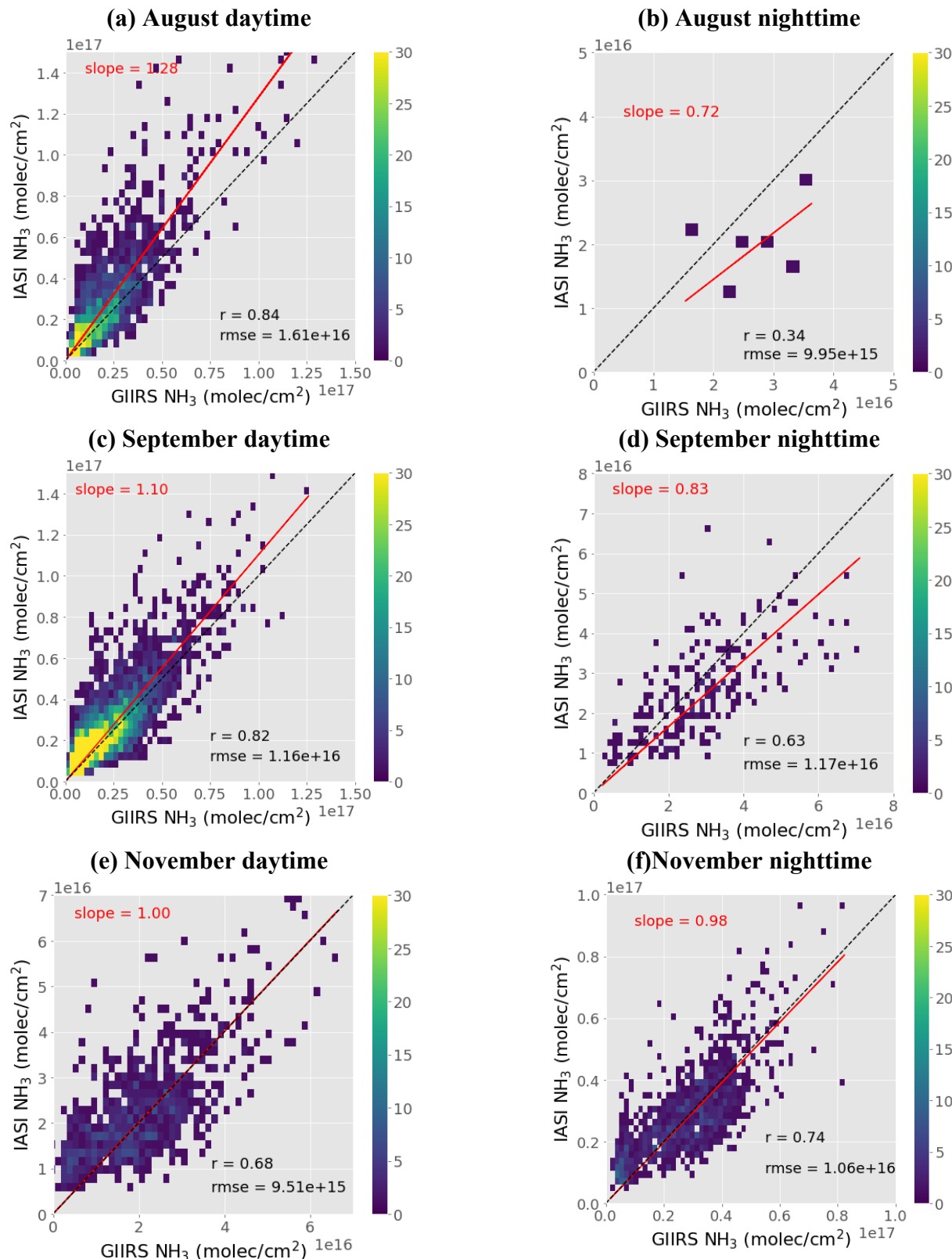

**Figure A3. Collocation intercomparison of NH₃ columns between GIIRS and IASI NH₃ retrieved columns, similar to Figure 12 but for August, September, and November.**


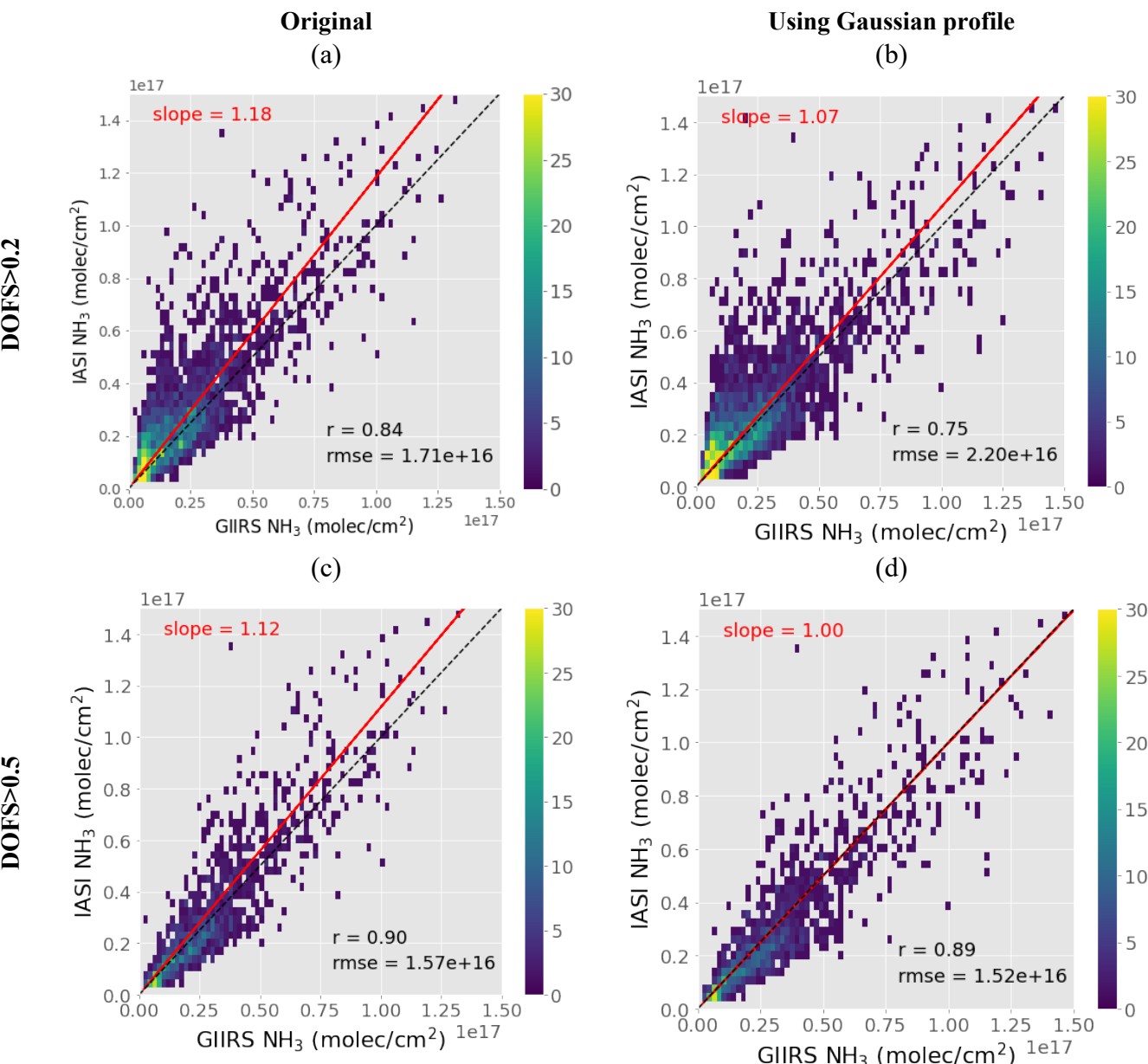

**Figure A4. Intercomparison of collocated GIIRS and IASI NH₃ columns for July daytime, similar to Figure 12 but for a new set of GIIRS NH₃ retrievals that use the Gaussian shape a priori profiles following the IASI v4 algorithm (Clarisse et al., 2023), while keeping the total column for the new a priori profile unchanged. The first and the second columns are the comparisons using the original and the new set of GIIRS NH₃ retrievals, respectively. The comparisons by two data filtering criteria based on DOFS>0.2 and DOFS >0.5 are shown in the first and second row, respectively.**