# Peer review of "Optimal estimation retrieval of tropospheric ammonia from the Geostationary Interferometric Infrared Sounder onboard FengYun-4B"

_Atmospheric Measurement Techniques, 2023_

## Author Comment (AC1)

Zeng et al. present NH3 retrieval results from measurements of the Geostationary Interferometric Infrared Sounder onboard FengYun-4B. Compared to previous work, their data product now allows for studies of the diurnal cycle of NH3, due to the geostationary orbit of the satellite. The topic of the manuscript is certainly within the scope of AMT and of scientific importance. However, before publication in AMT, major revisions are necessary, as described below.

We thank the reviewers for his/her constructive comments and suggestions to improve the quality and clarity of our manuscript. We have made major and careful modifications to the original manuscript according to all the comments and suggestions from the reviewers. The major changes include:
1. In the main manuscript, we restructured the manuscript and focused on the GIIRS $NH_3$ retrieval results using just the strong absorption micro-window, 955-975 $cm^{-1}$, and moved the comparison with retrieval results using 920-940 $cm^{-1}$ micro-window to the supplementary materials;
2. We have made detailed comparison with IASI's most updated version 4 $NH_3$ data products for the same period (July to December of 2022), in which we carried out spatial comparison and point-by-point collocation comparison to demonstrate the consistency of our retrievals with IASI data;
3. Throughout the revised manuscript, we have enlarged all figure fonts to increase readability.

Item-by-item responses to the specific comments are provided below, in which the reviews' comments are in blue, our responses in **black**, and modifications of the original manuscript are indicated by highlight in yellow in the revised manuscript.

**General points:**
- Labels and ticks are for most of the figures very small and hard to see without zooming the PDF to 200%. I would suggest to increase font size and line width considerably.

Throughout the revised manuscript, we have enlarged all figure fonts and line width of figures where necessary to increase readability.

- Until the beginning of Section 4 it was not clear to me, if one retrieval using both microwindows is presented here, or if two retrievals using one of the microwindows each are compared against each other. This should be stated much more explicitly in the beginning of the manuscript. It should be also motivated, why two retrieval approaches are compared against each other here. What should the reader learn from this exercise? Or are there also retrieval results shown using both microwindows (as indicated by the caption of Fig. 10)? I am very confused and I am not sure if I know which kind of retrieval was shown in which part of the manuscript.

In the main manuscript, we have re-structured the paper and focused on the GIIRS $NH_3$ retrieval results using just the strong absorption micro-window, 955-975 $cm^{-1}$, and moved the comparison with retrieval results using 920-940 $cm^{-1}$ micro-window to the supplementary materials (see **Supplementary Text S1 and Figures S4 and S5**). The motivation of this comparison is to investigate the consistency of the retrieval results using two different micro-windows, and to check if different window may give similar results, as a way to quantify the robustness of the spectra features. From the comparison, we do see high consistency in

results from the two micro-windows. We have added related statements in the revised manuscript and the supplementary materials.

- In Section 3.2, I miss general information about the retrieval. What quantities are derived from the retrieval? Is the NH3 concentration derived as a profile or only as a total column? What about temperature and the other trace gases? Until now, this section mainly collects references to theory. This is also important, but does not help to understand what kind of retrieval is performed in this work. Further, specific advantages and drawbacks of the selected retrieval approach should be at least mentioned here. Also the importance of the a priori profile in case of optimal estimation.

In the revised manuscript, we have added more details of the retrieval algorithm, including the state vector, profile retrieval of $NH_3$, and other interfering gases, please see **Sections 3.1** and **3.2**. We have added **Table 1** to show the parameters in the state vector to be retrieved from the retrieval algorithm.

**"The parameters in the state vector to be retrieved from the algorithm are listed in Table 1. Vertical profiles of $NH_3$ and $H_2O$ are retrieved, while for minor interference gases, total columns are retrieved by scaling an a priori profile. Other parameters to be retrieved include the surface skin temperature, a scaling factor for the atmospheric temperature profile, and the slope and curve for the surface emissivity. We only retrieve the layers below 200 hPa and use the a priori for layers above to compute total columns."**

Two widely used methods for retrieving $NH_3$ from space have been mentioned in the revised manuscript: "**A similar micro-window has been used to retrieve $NH_3$ using IASI (Clarisse et al., 2010) and GOSAT (Someya et al., 2020), while neural-network-based studies using hyperspectral radiance index (HRI) adopted a much wider window (e.g., Whitburn et al., 2016).**"

In addition, we have added the a priori covariance matrix and stressed the importance of the a priori profile in **Section 3.3** in the revised manuscript.

- In Section 4, the retrievals using different microwindows are compared. It is certainly of interest, which spectral region is suited best for the retrieval (or maybe both regions together?), but one important point is missing: There are certainly (temporally and spatially) co-located measurements by IASI, CrIS, …, which could be used to judge, which of the retrieval approaches better matches the established NH3 columns. I really think this kind of validation approach is missing here. Just saying that microwindow #2 has a stronger signal was already clear by looking at the spectral signature in Fig.2. So what is the point of the comparison of the different retrievals, if not comparing them to independent data?

Thanks for the great suggestion. It is important to carry out comparison with collocated measurements from independent instruments. In the revised manuscript, we have made such detailed comparison with IASI's most updated version 4 $NH_3$ data products for the same period (July to December of 2022). Dr. Lieven Clarisse and Dr. Martin Van Damme who developed the IASI NH3 retrieval algorithm have provided guidance on the comparison and they have joined as co-authors of this manuscript.

Two types of comparisons have been made: (1) Comparison of spatial distribution on representative days. The spatial distribution of $NH_3$ columns on two representative days: July 07, 2022 (summer, local daytime) when there was mostly positive TC, and December 18, 2022 (winter, local nighttime) when there was

mostly negative TC are compared between GIIRS and IASI; (2) Collocated point-by-point comparison. a spatially and temporally collocated point-by-point comparison between FY-4B/GIIRS and IASI $NH_3$ retrievals. We consider observations to be collocated when the distance between the centers of the pixels is less than 6km (half of the FY-4B/GIIRS footprint size) and the observation time difference less than 1 hour. In both cases, good agreements are found in general with fitted slopes close to unity.

In addition, we have also conducted an experiment to re-generate a new set of GIIRS $NH_3$ retrievals using the Gaussian a priori profile provided by IASI to explain the small difference between the two datasets in summer. The result from this experiment demonstrates that the systematic bias between GIIRS and IASI in Summer daytime is caused by the difference in their a priori profile structure.

Please refer to **Section 4.3** for the details and results of the comparisons.

- Section 5 seems to be out of place and the title "Discussion" is misleading. In my opinion, this section shows attempts to quantify the errors of the measurements. This is an important task, but it would be interesting to know the uncertainties of the measurements, before diurnal cycles are discussed in Section 4. I suggest to add both subsections 5.1 and 5.2 to section 3, since they also describe the retrieval.

As suggested, in the revised manuscript, we have moved Sections 5.1 and 5.2 to "**Section 3.5 Retrieval experiments for quantifying retrieval error.**"

- Some sources of errors are mentioned in the sections 5.1 and 5.2. Are there any other random or systematic error sources, that are expected to influence the retrieval? I guess the calibration of the spectra should also come with uncertainties, which should be considered for the error of the retrieval results.

We agree that the spectral calibration uncertainty is expected to influence the retrieval. As described in **Section 3.2**, if we just use the spectra noise as described by the NEdT from the GIIRS L1B data product, the averaged reduced $\chi^2$ value from the optimal estimation $NH_3$ retrieval is systematically larger than 1.0, suggesting the forward model error is not properly characterized by the spectra noise. So we have enlarged the spectra noise by a factor of 2.0. This extra noise represents the unaccounted uncertainty from the forward model and absorption spectroscopy by the original instrument noise alone, which may come from the spectral calibration uncertainty.

**Specific points: (**line numbers are given in the beginning of each point)

- 14: "our retrievals are implemented using two different absorption micro-windows": For me this sounds like both microwindows are used in the same retrieval. The following sentences sound more like two retrievals based on the two microwindows are performed and compared. Please clarify. (see also general comment)

In the revised manuscript, we have re-structured the manuscript and focused on the retrieval results from 955-975 cm$^{-1}$ micro-window only and moved the comparison using two micro-windows to the supplementary materials.

- 17: Typo: no -> No

Changed.

- 28-36: In this introduction to NH3, it would be also helpful to mention typical atmospheric loss processes of NH3.

In the revised manuscript, to mention the loss process of $NH_3$, we added "**In addition, ammonia reacts with acids (e.g., $H_2SO_4$, $HNO_3$) and produces ammonium containing aerosols that degrade air quality (Seinfeld and Pandis, 2006)**."

- 37-44: For completeness, it would be also worthwhile to mention that NH3 has been measured in the upper troposphere of the Asian Monsoon by infrared limb sounders.

In the revised manuscript, we added "**In addition, $NH_3$ has been detected in the Asian summer monsoon upper troposphere using the emission spectra from the infrared limb sounder Michelson Interferometer for Passive Atmospheric Sounding (MIPAS; Höpfner et al., 2016)**."

The following reference has been added:
Höpfner, M., Volkamer, R., Grabowski, U., Grutter, M., Orphal, J., Stiller, G., von Clarmann, T., and Wetzel, G.: First detection of ammonia (NH3) in the Asian summer monsoon upper troposphere, Atmos. Chem. Phys., 16, 14357–14369, https://doi.org/10.5194/acp-16-14357-2016, 2016.

- 70: Suggestion: "developed by the group (Zeng et al., 2022) based" -> "developed by Zeng et al., (2022), based"

Changed as suggested.

- 70: Typo: optical -> optimal

Changed.

- 71: absorption feature -> absorption features

Changed.

- 72: The sentence "The primary goals are ..." feels out of place there. It is between two sentences explaining the spectral properties of the retrieval. The sentence could be moved to the beginning of the paragraph (before the sentence "The retrieval algorithm uses ...").

Changed as suggested.

- 74: "our retrievals are ...": It is still not clear to me, if the authors are explaining a single retrieval setup using both microwindows (in this case, this phrase should be used in

singular "our reatrieval is ..."), or if two different retrieval approaches with different microwindows are compared (in this case, this should be written much more specific).

In the main manuscript, we changed to use just the strong absorption micro-window, 955-975 cm$^{-1}$, and moved the comparison with retrieval results using 920-940 cm$^{-1}$ micro-window to the supplementary materials

- 75: "their difference in retrievals": Same comment as for line 74.

Please see our response above.

- 90: The introduction of Fig. 1a is misleading. It should be clearly stated in the text that this panel shows NH3 inventory data and no measurements!

We added "…, **as indicated by the bottom-up inventory map** ..." to explicitly state that the data are bottom-up inventories.

- 93: "GIIRS is in principle capable of measuring trace gases": Which other trace gases than NH3 have been retrieved or are planned to be retrieved in the near future?

We added "…, **including NH$_3$ and carbon monoxide (Zeng et al., 2022),** …" in the revised manuscript.

- 99: "comparable to existing infrared sounders": I suggest to give some examples of NedT for infrared sounders, which also measure NH3 columns.

In the revised manuscript, we added "**The low instrument noise for FY-4B/GIIRS, comparable to existing infrared sounders (e.g., ~0.2K@280K for IASI and ~0.04K@280K for CrIS; Van Damme et al., 2014; Shephard et al., 2020), makes it possible to accurately retrieve NH$_3$ over East Asia.**"

- 100: I do not understand the formulation "provide strong constrain" in this context, but I am no native speaker.

We have rephrased it to "**makes it possible to accurately retrieve NH$_3$ over East Asia.**"

- Figure 1c: It would be helpful to zoom into the used microwindows (in an additional panel?) and highlight the spectral features of NH3, which are used for the retrieval.

A sub-figure has been added to Figure 1(c) that zooms into the used micro-window. In the caption of Figure 1, we added "**The micro-window for NH$_3$ retrieval is also shown in red rectangle and as inset. The NH$_3$ absorption features are shown in Figure 2.**"

- 124: It would be good to also mention the possible interfering species and discuss a bit the content of Figure 2.

We added "**The micro-window contains strong NH$_3$ absorption features that are distinguishable from important interference gases (CO$_2$, O$_3$ and H$_2$O) in the absorption window, as demonstrated**

**in Figure 2 from a sensitivity experiment that compares absorptions of NH$_3$ and perturbed interference gases.**"

- 128: Please define the acronyms ECMWF and ERA5

Defined.

- 130: Pleas define the acronym CAMS. Are the concentrations or the fluxes used from this data set? Why did the authors decide for this specific CAMS data set? Why are other trace gas data taken from ERA5 instead?

The acronym CAMS has been defined. The concentrations are used from this dataset. The ERA5 is primarily used for gases (H$_2$O and O$_3$) because it has a higher temporal resolution (1-hour). For gases that are not available from ERA5, we extract the data (CO$_2$) from CAMS instead.

- Figure 2: N2O is mentioned to be considered for the radiative transfer model, but there are no spectral features shown in this figure. Why?

We have removed N$_2$O as an interference gas since it has no significant contribution in this spectral region.

- 134: "Since the NH3 is more ...": It seems like the retrieval grid is introduced and motivated here. I do not understand, why this part starts with a comparison to CO. Maybe, the retrieval is compared to previous work of the authors? I would suggest to start a new paragraph here and then explain, how the retrieval grid is chosen for the NH3 retrieval, motivate the choice and then compare it to similar retrieval setups with the same instrument. I am also missing the information, if the retrieval is on pressure or altitude grid, since some numbers are given in pressure, others in kilometers.

In the revised manuscript, we have started a new paragraph and rephrased the sentences to include all necessary information:

**"Since NH$_3$ is short-lived and highly concentrated in the PBL, we therefore only retrieve the layers below 200 hPa. The forward model uses fixed vertical grids with equally separated layers with similar thickness (about 1 km for layers below 200 hPa and about 5 km for layers above), which is close to the grid settings in Hurtmans et al. (2012) and Clough et al. (2005). The thickness of the bottom layer is variable and determined by the surface pressure of a specific location. The number of layers below 200 hPa ranges from 7 (for high altitude regions such as the Tibet Plateau) to 11 layers (for low altitude regions such as the ocean)."**

The retrieval is on pressure grid, as described in the rephrased paragraph.

- 136: "The number of layers for retrieval ranges from 7 (at high altitude) to 11 layers (at low altitude).": I do not understand this sentence. Maybe it becomes clearer, if the altitude grid itself is introduced properly before.

We have rephrased the sentence to: "**The number of layers below 200 hPa ranges from 7 (for high altitude regions such as the Tibet Plateau) to 11 layers (for low altitude regions such as the ocean)**"

The retrieval grids are also shown in Figure 3(a).

- 147: "The auxiliary parameters include ...": I do not understand the meaning of "auxiliary parameters" in this context. Does that mean that all of these quantities are used by the forward model? Or are some of these parameters also fitted? What exactly are "scale factors"? Are these factors adjusted by the fit?

In the revised manuscript, we have rephrased the descriptions to:

"**The parameters in the state vector to be retrieved from the algorithm are listed in Table 1. Vertical profiles of $NH_3$ and $H_2O$ are retrieved, while for minor interference gases, total columns are retrieved by scaling an a priori profile. Other parameters to be retrieved include the surface skin temperature, a scaling factor for the atmospheric temperature profile, and the slope and curve for the surface emissivity. We only retrieve the layers below 200 hPa and use the a priori for layers above to compute total columns.**"

The scaling factor for the atmospheric temperature profile is adjusted in the spectral fitting process.

- 149: "...which maximize the a posteriori probability given the FY-4B/GIIRS spectra." I do not understand this formulation. Please rephrase.

We have rephrased it to: "**which minimizes the spectral fitting error.**"

- 157: A_ij has not been introduced until here. Suggestion: "where each element A_ij of A represents ..."

Changed as suggested.

- 160: Suggestion: "by 2.0 times" -> "by a factor of 2.0"

Changed as suggested.

- 175: "(2) it is not ...": I am not sure, if I understand this correctly. Are the authors trying to say that they would need to rely on model profiles (which may have an incorrect diurnal cycle), if they want to use time dependent a priori profiles?

We have rephrased it to: "**(2) it is not applicable to get a reasonable a priori estimate for all hours in a day from just the spectra (e.g., using channel brightness temperature difference as in Shephard et al. (2011) and Warner et al. (2016)) without relying on model simulations, …**"

- 178 and following: I am not sure, if I understood this correctly: Only one profile is used for all of the retrievals. This profile is an average over all land regions in the given lat/lon boxes for 2022. Please try to formulate more precisely here!

We have rephrased the related statement to: "**The single a priori NH₃ profile, as shown in Figure 3(a), for all retrievals in the retrieval algorithm is derived from NH₃ simulations …**"

- 180: Since the sentence before was discussing a model, it should be "One year of simulation" instead of "One year of measurements".

Changed. It should be "One year of simulation".

- Figure 3: The error bars of the average NH3 profile reach negative mixing ratios. How is that possible for model simulation results? Is the profile averaged using the mean of all profiles? Or the median? Or something different? Further, which kind of mixing ratio is shown here? Volume mixing ratio? Mass mixing ratio? Please add!

The following statement has been added to the revised manuscript to introduce these in more details: "**The negative value toward the lower end of the error bar does not have physical meaning, it is caused by the large standard deviation derived from model simulations that do not strictly follow a normal distribution. The a priori total NH₃ column is about $1.5 \times 10^{16}$ molecules/cm². To construct the correlation matrix, we used a correlation length of 3 km based on our analysis of the GEOS-CF reanalysis. Most of the layers show correlation lengths between 1 to 3 km and we use upper bound (3 km) to increase the stability of the retrieval system. The covariance matrix calculated based on the a priori error and the correlation matrix is shown in Figure 3(b).**"

This is volume mixing ratio. The x-label has been changed.

- Section 3.4: I would expect to perform cloud filtering before doing the retrieval to avoid unnecessary fits of cloud contaminated spectra. I think the subsection headline "post-filtering" is not adequate for the cloud filtering part then. Further, I would expect this part earlier in section 3, since it is performed before the retrieval itself.

Thanks for pointing this out. In the revised manuscript, we have moved the cloud screening part to the end of Section 2.

- Figure S1: It would be more interesting to see the histograms before applying the filters and with an enlarged x-axis.

We have added the histogram before applying the filtering in the **Supplementary Figure S2**.

- 199: Suggestion: "suggesting satisfying goodness of fit." -> "suggesting good fit quality."

We have changed it to: "**suggesting a satisfactory goodness of fit**".

- 200: I do not understand the formulation "do not have enough constrain from the observed spectra" in this context. Please rephrase.

We have rephrased it to: "**In the following analysis, an extra filter based on DOFS may apply to exclude data with low DOFS and, therefore, low information content extracted from the observed spectra.**"

- 215: "Fortunately, no large systematic bias is observed ...": I think, there is a banana-shape feature of the correlation (but I may be wrong, see my comment on Figure 4). If there is this kind of banana-shape, what does this mean for the retrieval performance? How do you know, which microwindow selection is better? As already mentioned in the general points, I think such correlations are more helpful against independent measurements, e.g. from IASI, CrIS, ...

In the revised manuscript, the comparison of $NH_3$ between two micro-windows has been moved to the **Supplementary Figure S5**. One-to-one lines have been added in the updated figures.

The scatter plot appears to be a banana-shape mainly because of the saturated color scheme used for the data points. In the revised manuscript, the figure has been re-plotted using a color bar with log10 scale. We have also added an extra figure with DOFS>0.7 to demonstrate how the agreement may improve with retrievals with high DOFS. The revised figure shows that the two datasets agree very well, and there is no such "banana" shape in the data. The data agreement can also be seen in the histogram figures of $NH_3$ difference, that shows an even distribution without systematic bias. For the comparison with DOFS>0.7, we see that the agreement improves. The comparison suggests that the two retrievals agree well with each other, and no significant bias exist, especially for retrieval with high DOFS.

In the revised manuscript, we restructured the paper and focused on the 955-975 cm$^{-1}$ micro-window only, which shows a higher DOFS and has been widely adopted by several other retrieval data products. In addition, we have made detailed comparison with IASI's most updated version 4 $NH_3$ data products for the same period (July to December of 2022). Please refer to **Section 4.5** for the details of comparison.

[Figure]

**Part of Figure S5. (b) the retrieved $NH_3$ columns filtered by DOFS>0.5. For the comparison of columns, in total, 1.1 million data points are available. The correlation coefficient between the two column datasets is 0.82 with a root-mean-square-error of 9.2×10$^{15}$ molec/cm$^2$. The histogram is also shown; (c) the retrieved $NH_3$ columns filtered by DOFS>0.7. In total, 0.5 million data points are available for comparison. The correlation coefficient is 0.86 with a root-mean-square-error of 8.6×10$^{15}$ molec/cm$^2$. The mean errors are 2.0×10$^{15}$ and 2.7×10$^{15}$ molec/cm$^2$, respectively, for (b) and (c).**

Figure 4: I think, a 1:1-line would be very helpful here to guide the eye. For me it looks like there is a banana-shape in both correlations, but this is not discussed at all in the text. Maybe my eye is wrong here, but a 1:1-line would help!

In the revised manuscript, the comparison of $NH_3$ between two micro-windows has been moved to the **Supplementary Figure S5**. One-to-one lines have been added in the updated figures. Please see our responses to your last comment.

- 229: I suggest to replace "non-source" by "background"

Changed as suggested.

- 239: "A typical "butterfly" shape can be seen in almost all cases." What about the other cases, e.g. North-India in July? A short comment would be helpful.

In North India in July, there was no enough data are available. Short comment "**except for North India in July when there are much less observations due to clouds**" has been added to the revised manuscript.

- 244-251: Fig. S2 shows very large TC for the Tibetan Plateau for BJT 14-15 h. However, the DOFS are very low here (almost zero). How does this agree with the mentioned "strong correlations between DOFS and TC"?

As we explain in the text, the DOFS is a function of both TC and $NH_3$ abundance. In Tibetan Plateau although it has large TC, the DOFS is low because of its very small $NH_3$ columns.

Related statement has been rephrased in the revised manuscript:
"**This strong correlations between DOFS and TC or $NH_3$ abundance is also reflected in the spatial maps of DOFSs …**" and "**The source regions in North China Plain and North India have higher DOFSs than other non-source regions, such as Tibetan Plateau although it has large TC**."

- 256: "We can see the diurnal AK values ...": First, the quantity "averaged averaging kernel row" should be introduced and what we can learn from it. Further, in Fig. 7 the x-axis-label should be corrected to "averaged averaging kernel row". I also miss a altitude-resolved averaging kernel plotted for different altitudes without averaging it, at least for an example in the supplement.

We have changed it to "Averaged averaging kernel diagonal vectors", which are measures of the DOFS for each vertical layer. The x-axis label has also been changed accordingly. We have rephrased the statement to **"… using the corresponding averaging kernel diagonal vectors, which are measures of the DOFS for each vertical layer."**

In addition, an example of an altitude-resolved averaging kernel plotted for different altitudes without averaging it is shown in the **Appendix Figure A1**.

- 293: "The data gaps ...": I would have expected that cloud filtering would also considerably impact data availability, in particular in North India during the Monsoon season. In fact, in Fig.

10, no time series can be shown for North India in July-August. I miss a comment for this at all in Section 4.3.

Thanks for your suggestion. In the revised manuscript, we added the following statement:
**"The cloud filtering has considerably impacted data availability, in particular in North India during the Monsoon season in July-August."**

- 297: "Interestingly, ...": Comparing the steps of the diurnal cycle of different months in Figures 8 and 9 with the half-year average inventory in Fig. 1a seems not like a fair comparison. On the one side, diurnally- and seasonally-resolved satellite measurements with data gaps in North India (a region specifically mentioned in the comparison) are compared to a half-year average. It's apples and oranges. Further, in July-August, there are times, in which North China Plain shows higher NH3 columns than North India.

In the revised manuscript, we have removed the comparison with bottom-up inventories, and rephrase the statement to be: **"As explained in Wang et al. (2020), the causes of high NH$_3$ loading in North India, which are slightly different from that in the North China Plain, are due to the weak chemical loss and weak horizontal diffusion in North India."**

- 307: Typo: capture -> captured

Changed.

- 308: "In addition, ...": To which months does this sentence refer to? Further, would more precipitation also wash-out the highly water soluble NH3 from the atmosphere?

It is referring to the summer months. We have rephrased the sentence to "In addition, the relatively low temperature and higher humidity **in the nighttime, relative to the daytime**, contribute to the conversion from NH$_3$ to particulates that leads to a lower NH$_3$ concentration."

The removal of NH$_3$ from the atmosphere is call wet deposition. It is an important mechanism of NH$_3$ loss. However, it is not the main reason for the diurnal cycle as discussed in this paragraph.

- Section 4.3: This section should be restructured. In the first paragraph, the diurnal cycles shown in Fig. 10 are already briefly introduced, then the next paragraph starts with some explanations of the observed diurnal cycles, while in the end of the section, the diurnal cycles are again introduced again, but in more detail. I suggest to start a new paragraph for the description and discussion of Fig. 10. In this paragraph, please first describe, what can be seen in the figure, and then explain, why it makes sense, what we see.

We have re-structured the section (now **Section 4.2**) following your suggestions. The explanation of the observed diurnal cycles has been rephrased as:

**"The general diurnal cycle of NH$_3$ columns can be primarily explained by three possible driving factors, as concluded in the summary in Clarisse et al. (2021), including the day-night difference in**

agriculture activities as a major source of NH₃, the temperature dependence of NH₃ emissions driven by diurnal and seasonal temperature changes, and the conversion between NH₃ gas and particulate driven by the day-night change of meteorological conditions. These can be used to interpret the quantitative analysis of the diurnal cycle as shown in Figure 10 for North China Plain and North India.”

Please see Section 4.2 in the revised manuscript for the details.

- Figure 8/9: I do not understand, why the order of the panels is different compared to Fig. 5. It would be easier, if this would be consistent throughout the manuscript. E.g. keep it in a way that the diurnal variation is visible within a row and the seasonal changes are visible within a column. Maybe it would be best to change Fig. 5 (and S2).

We have changed the structure of **Figures 8** and **9** such that, to be consistent throughout the paper, the rows represent different months.

-339: "The error bar represents one standard deviation.": So the error bars rather show variability within the region than an error/uncertainty of the measurement? Since this variability is quite large compared to the observed diurnal cycle, this variability should be discussed in Section 4.3 somewhere.

We added the following statements in the revised manuscript:
**“Note that the variability of NH₃ columns within the region is large as shown by the error bars (one standard deviation). This large variability is a result of NH₃'s short life time and the spatial heterogeneity of its emissions.”**

- Section 5.1: In the end of the section, a noise error is estimated for the retrieval. It would be interesting, how this error is for the diurnal cycles shown in Fig. 10 compared to the signal of the diurnal cycle itself. Is the error larger or smaller than the variability (which is shown in Fig.10 with error bars)? Is the amplitude of the diurnal cycle larger than the scaled random error from this exercise?

To compare the uncertainty of the retrievals with the diurnal variabilities, we added the following statements: **“Moreover, when compared with the averaged uncertainty of a single retrieval ($1.37×10^{16}$ to $1.67×10^{16}$ molec/cm² as derived from the retrieval experiment in Section 3.5), the day-night contrast of the averaged diurnal variations of NH₃ columns as shown in Figure 10 may not be significant for the North China Plain in September-October and the North India in November-December.”**

- Figure 11: For me, it looks like there is a low bias for the NH3 retrieval: There are considerably more dots far away from the 1:1 line in the lower-right half of the plot than in the upper-left part. And a non-negligible number of the extreme outlier have a DOFS > 0.5. This should be discussed in the text. Do the authors have any idea about the source of this systematic error?

The data points on the lower-right half are mostly retrievals without high information content (low DOFS). As a result, these retrievals are close to the a priori value, which is about $1.5×10^{16}$ molecules/cm².

For those outliers that have high DOFS but with poor agreement, they are likely caused by large difference of the real NH$_3$ profile and the a priori profile. Because the satellite retrieval cannot resolve all layers, the extrapolation of the retrieval profile scaling may lead to large bias if the truth NH$_3$ profile is far away from the a priori profile. Related statements have been added to the revised manuscript.

- 385: I think, the reference should be to Section 5.1

Changed.

- 386: In Section 5.1, also absolute uncertainties are given. That would be also helpful here.

Added. The absolute root-mean-square errors for both cases are about the same $3.4\times10^{15}$ molec/cm$^2$.

Figure 12 b/c: A 1:1-line would be helpful here. Further, it seems like the correlation splits into "branches", in particular for Fig. 12b. So, the correlation does not evenly scatter around the 1:1 line. This should be discussed in the text. Are there any explanation for this behavior?

The Figure has been moved to the **Supplementary Figure S6**. 1:1 line has been added to (b) and (c). As you mentioned, the correlation does not scatter along the 1:1 line. For (b) with retrieved NH$_3$ column using the reduced PBL excess profile, the retrievals are underestimated compared with the original retrievals. For (c) with retrieved NH$_3$ column using the enhanced PBL excess profile, the retrievals are overestimated compared with the original retrievals. These differences are mainly caused by the a priori total columns.

- 429: The given link only leading to the FY-4B/GIIRS CO data, but no NH3 data is available there. Please add the missing data to the website, or give the correct link. For this and all other mentioned data resources, it would be further better to have the data versioned and tagged with a DOI.

The associated NH$_3$ datasets and a data user guide have been uploaded to the PKU Opendata repository (https://opendata.pku.edu.cn/dataverse/FYGEOAIR). The designated DOI for this dataset is: https://doi.org/10.18170/DVN/VJ4MLO.

---

## Author Comment (AC2)

**Reviewer 2**

The paper by Zeng et al., presents a NH3 retrieval product from the GIIRS infrared sounder on board FY-4B. Geostationary NH3 measurements are novel and of high interest to the scientific community.  The paper is not bad, but there is certainly a lot of room for improvement, both in content, form and depth, which is why I would recommend a major revision, after which the paper needs to re-reviewed. I would also encourage the authors not to see the revision as a hurdle that needs to be overcome to publish, but rather as an opportunity to improve the paper, and to increase its impact.

We thank the reviewers for his/her constructive comments and suggestions to improve the quality and clarity of our manuscript. We have made major and careful modifications to the original manuscript according to all the comments and suggestions from the reviewers. The major changes include:

1. In the main manuscript, we restructured the manuscript and focused on the GIIRS NH$_3$ retrieval results using just the strong absorption micro-window, 955-975 cm$^{-1}$, and moved the comparison with retrieval results using 920-940 cm$^{-1}$ micro-window to the supplementary materials;
2. We have made detailed comparison with IASI's most updated version 4 NH$_3$ data products for the same period (July to December of 2022), in which we carried out spatial comparison and point-by-point collocation comparison to demonstrate the consistency of our retrievals with IASI data;
3. Throughout the revised manuscript, we have enlarged all figure fonts to increase readability.

Item-by-item responses to the specific comments are provided below, in which the reviews' comments are in blue, our responses in **black**, and modifications of the original manuscript are indicated by highlight in yellow in the revised manuscript.

A. Major Comments
A.1 General comments on the figures
 - As a general rule, the font size of the figures should match that of the text, or slightly smaller.  A lot of text is currently simply unreadable on print-out.

Throughout the revised manuscript, we have enlarged all figure fonts to increase readability.

 - Again, as a general rule, try to decide whether a figure should be a one-column figure or a two column-figure in the final manuscript, and then utilise exactly the full width or exactly half of the full width of the text. This applies at least to Fig 1C (bottom row could be full page width), Fig 2, Fig 4, Fig 6, Fig 7 and Fig 10.

We have changed the size of these figures. **Figure 1(c)** has been changed to full page width. **Figures 2, 4, 6, 7 and 10** have been enlarged to full page width.

 - For large multipanel figures with the same x and y axis, the axis tick labels and axis labels can be removed from the middle panels. This in effect increases drastically the space available for the actual figures. For instance in Figure 8, the degrees East can removed on top, between the first and second row, and the second and third row (in this way only keep it at the bottom). Likewise, the degrees North can be removed between the first and second column and between the second and third column, and only kept on the left side.

After this, all panels can be enlarged. This applies to Figure 5, Figure 6, Figure 8, Figure 9 and Figure S2.  For Figure 5, one colourbar suffices, which can go to the bottom, again allowing for larger and more visible panels.
 - Wherever possible, please export figures in a vector format (https://en.wikipedia.org/wiki/Vector_graphics, e.g. pdf), to avoid pixelation of lines and text, especially visible when zooming in on screen. For instance figure 10 or 12 (but ideally for all figures).

In the revised manuscript, we have removed the repeated x- or y-tick labels in **Figure 5, 8, 9, and S2**, and enlarged the figures using the extra space. We have remade the figures to use vectorized format where necessary to increase readability.

**A.2: Other major comments**
 - 1 km layering at the bottom of the atmosphere is not sufficient (at least 500 m is required below 2 km). 1 km is not sufficient for modelling fast varying temperature (e.g. thermal inversions at night), nor for representing the fast NH3 (especially at night) and H2O variations (always). I can confidently say that the fit residuals will decrease once a finer layering is adopted (even just for the H2O fit). The fact also that there was a need to multiply the spectral noise by two, gives a hint that the fits can and should be improved. I realize that redoing all the fits is a major undertaking, but worth it, if the product is going to be further extended in time and used by the community. If a coarse spectral resolution emissivity database was used, I would also encourage to look for better alternatives.

We have tested using the suggested a priori with 0.5km thickness layers for bottom 2km and re-run all retrievals on 2 representative days (July 07 and December 18 of 2022). Theoretically, as the reviewer pointed out, that this would significantly improve the spectral fitting. This is given that the spectra contain enough information to resolve the $NH_3$ vertical structure in the bottom atmosphere. However, the spectra as in our retrieval can have less than one degree of freedom, suggesting the spectra is not able to correctly resolve lower atmospheric structures, similar to most existing infrared sounders. This can be seen from the comparison of the reduced $\chi^2$ and RMSE of spectral fitting errors from both retrievals (please see the following figure). The reduced $\chi^2$ and the RMSEs of spectral fitting errors from the updated retrievals are basically the same as the original results, suggesting the use of 0.5km thickness layers for bottom 2km may not improve the spectral fit. We therefore continue to use the current vertical grid settings.

For the spectra noise that has been multiplied by 2.0, this is more likely due to the uncertainty in the forward model (such as the used ABSCO lookup table), which is close to random, not systematic.

For the emissivity database, over the narrow micro-window in our retrieval, it is sufficient to assume that the emissivity does change significantly. In our retrieval, we have fit a linear trend and a curve to the emissivity to allow it to change as a function of wavenumber.

We have added relate statements to the revised manuscript.

[Figure]

Figure. Comparison of reduced $\chi^2$ and spectral fitting error between original retrievals and retrievals using a priori $NH_3$ profile with 0.5km thickness layers for bottom 2 km on two representative days (July 07 and December 18 of 2022).

 - Section 3.1 "forward model" already introduces elements of the retrieval. It makes it confusing and not well structured. E.g. the layering used for the atmosphere doesn't necessarily have to match the layering used for the retrieval.

You are right that the layers for the retrieval algorithm can be different from those in the RT model. In the revised manuscript, we have added details about the atmospheric layers used in the forward RT model and for the retrieval algorithm. The rephrased statements are:
**"Since the $NH_3$ is short-lived and highly concentrated in the PBL, we therefore only retrieve the layers below 200 hPa. The forward model uses fixed vertical grids with equally separated layers with similar thickness (about 1 km for layers below 200 hPa and about 5 km for layers above), which is close to the grid settings in Hurtmans et al. (2012) and Clough et al. (2005). The thickness of the bottom layer is variable and determined by the surface pressure of a specific location. The number of layers below 200 hPa ranges from 7 (for high altitude regions such as the Tibet Plateau) to 11 layers (for low altitude regions such as the ocean)."**

 - Section 3.2 I think a comprehensive table with all the forward/retrieval parameters would go a long way in making this section more understandable, and the paper more reproducible. For an AMT paper it is essential that these important technical aspects are

fully transparent. E.g. a table with 1st column: Parameter name, 2nd column, levelling used in forward model, 3rd column: levelling used for the retrieval, 4th column: a priori, 5th column covariance matrix (i.e. % variability if diagonal, and description of off-diagonal elements, e.g. correlation length). This table should at least include all parameters that are retrieved, but also the important atmospheric and surface terms (temperature/pressure)

Thanks for your great suggestion. Part of Section 3.1 has been rephrased to comprehensively introduce the input variables necessary to drive the RT model. In addition, we have added Table 1 that includes descriptions of all parameters in the state vector to be retrieved from the retrieval algorithm. The variable names, no. of variables, a priori values, a priori uncertainty, and necessary descriptions are included. The following statements have been added to Section 3.2:

**"The parameters in the state vector to be retrieved from the algorithm are listed in Table 1. Vertical profiles of $NH_3$ and $H_2O$ are retrieved, while for minor interference gases, total columns are retrieved by scaling an a priori profile. Other parameters to be retrieved include the surface skin temperature, a scaling factor for the atmospheric temperature profile, and the slope and curve for the surface emissivity. We only retrieve the layers below 200 hPa and use the a priori for layers above to compute total columns."**

- Figure 3: please provide an additional panel with the a priori covariance matrix. Showing the a prior NH3 profile is only showing half of the story. How was it calculated? Also from model output? Please discuss this matrix at some length, as the choice of covariance matrix is absolutely key (both diagonal and off-diagonal elements).

The a priori profile and the a priori covariance matrix are important inputs in the optimal estimation-based retrieval algorithm. In the revised manuscript, we added descriptions of the details of the used a priori and the covariance matrix. The following statements have been added or rephrased:

**"The single a priori $NH_3$ profile, as shown in Figure 3(a), for all retrievals in the retrieval algorithm is derived from $NH_3$ simulations from the Goddard Earth Observing System composition forecast (GEOS-CF; Keller et al., 2021) model developed by NASA's Global Modeling and Assimilation Office (GMAO). One year of simulation in 2022 is used to get the mean and standard deviation of $NH_3$ vertical distribution. To avoid over sampling of the background regions, only simulations in the representative land regions in east Asia (20°-60°N and 110°-120°E) and south Asia (20°-40°N and 70°-100°E) are used. The negative value toward the lower end of the error bar does not have physical meaning, it is caused by the large standard deviation derived from model simulations that do not strictly follow a normal distribution. The a priori total $NH_3$ column is about $1.5 \times 10^{16}$ molecules/cm$^2$. To construct the correlation matrix, we used a correlation length of 3 km based on our analysis of the GEOS-CF reanalysis. Most of the layers show correlation lengths between 1 to 3 km and we use upper bound (3 km) to increase the stability of the retrieval system. The covariance matrix calculated based on the a priori error and the correlation matrix is shown in Figure 3(b)."**

- Section 4: comparing two retrievals is very interesting, but obviously also opens the door to a lot of questions. Currently the reader remains rather unsatisfied.

In the main manuscript, since the main focus of this paper is the development of an $NH_3$ retrieval algorithm for FY-4B/GIIRS, we have re-structured the paper and focused on the GIIRS $NH_3$ retrieval results using just the strong absorption micro-window, 955-975 cm$^{-1}$, and moved the comparison with retrieval results using 920-940 cm$^{-1}$ micro-window to the supplementary materials (see **Supplementary Text S1** and **Figures S4 and S5**).

* For the column retrievals, would it not be better to apply averaging kernels in the comparison, to remove the impact of the a priori all together (and to what extend it is used by both retrievals). Perhaps both could be shown (with and without AVKs applied). I am not sure what is the best approach (which averaging kernel to apply to what), but there is certainly literature out there to guide you.

The motivation of this comparison is to investigate the consistency of the retrieval results using two different micro-windows, and to check if different window may give similar results, as a way to quantify the robustness of the spectra features. From the comparison, we do see high consistency in the results from the two micro-windows. The small difference between the two micro-windows may be traced back to the slightly different sensitivity. We have added related statements in the revised manuscript and the supplementary materials.

Since the main focus of this paper is the development of an NH3 retrieval algorithm for FY-4B/GIIRS, we did not apply the AK correction, but instead mentioned it in the supplementary text as a potential cause of the difference between the two retrievals.

* Figure 4b is misleading, as it is highly saturated with the colourbar as-is. I would propose to use a colourbar with log scale (e.g. 1 to 10000 or higher). Also, the vast majority of observations have a column below 5 10^16, and for these points, there seems to be a clear bias between the two retrievals, where the micro-window 1 is low-biased. I disagree strongly that "no large systematic bias is observed" (perhaps non-surprisingly because of the lower information content - but in that case applying averaging kernels would show this, and the bias should disappear). I would not show observations above 1e17, to allow higher level of detail for the low columns.

In the revised manuscript, the comparison of $NH_3$ between two micro-windows has been moved to the **Supplementary Figure S5**. One-to-one lines have been added in the updated figures.

Following your suggestion, we plot the scatter plots using a colourbar with log scale. The scatter plot appears to be a banana-shape mainly because of the saturated color scheme used for the data points. In the revised manuscript, the figure has been re-plotted using a color bar with log10 scale. We have also added an extra figure with DOFS>0.7 to demonstrate how the agreement may improve with retrievals with high DOFS. The revised figure shows that the two datasets agree very well, and there is no significant "banana" shape in the data. The data agreement can also be seen in the histogram figures of $NH_3$ difference, that shows an even distribution without systematic bias. For the comparison with DOFS>0.7, we see that the agreement improves. The comparison suggests that the two retrievals agree well with each other, and no significant bias exist, especially for retrieval with high DOFS.

In the revised manuscript, we restructured the paper and focused on the 955-975 cm$^{-1}$ micro-window only, which shows a higher DOFS and has been widely adopted by several other retrieval data products. We have made detailed comparison with IASI's most updated version 4 $NH_3$ data products for the same period (July to December of 2022). Please refer to **Section 4.5** for the details of comparison.

[Figure]

**Figure S5. Comparison of NH₃ retrievals using micro-window #1 (920-940 cm⁻¹) and micro-window #2 (955-975 cm⁻¹): (a) the DOFS, which shows a correlation coefficient of 0.97 for a total of 11.7 million data points; (b) the retrieved NH₃ columns filtered by DOFS>0.5. For the comparison of columns, in total, 1.1 million data points are available. The correlation coefficient between the two column datasets is 0.82 with a root-mean-square-error of 9.2×10¹⁵ molec/cm². The histogram is also shown; (c) the retrieved NH₃ columns filtered by DOFS>0.7. In total, 0.5 million data points are available for comparison. The correlation coefficient is 0.86 with a root-mean-square-error of 8.6×10¹⁵ molec/cm². The mean errors are 2.0×10¹⁵ and 2.7×10¹⁵ molec/cm², respectively, for (b) and (c).**

* For the comparison, it would be good to add a figure showing the mean residual (calculated-observed as a spectrum) for one or two selected regions, for both retrievals. For a well-behaved retrieval, there should be no systematic features, and noise should average out.

The following figure shows the spectral fitting residual in brightness temperature averaged over all post-screened retrievals in July 2022 for micro-window #1 in (a) and micro-window #2 in (b). The error bars represent two standard deviations of the fitting errors. The corresponding histograms of the fitting errors for all channels are shown on the right. For micro-window #1, the mean and standard deviation are -0.0027 K and 0.14 K, respectively. For micro-window #2, they are 0.0090 K and 0.15 K, respectively.

The spectral fitting errors in brightness temperature (BT) are small but show systematic patterns, that are persistent among observations at different hours, can be seen from the averaged fitting residual from all spectra. These patterns are likely caused by the uncertainty in molecular absorption properties in the

ABSCO lookup tables. Fortunately, these patterns are not correlated with the absorption feature of the target gas $NH_3$ and are not expected to significantly affect the retrievals of $NH_3$.

[Figure]

Figure. The spectral fitting residual in brightness temperature averaged over all post-screened retrievals in July 2022 for micro-window #1 in (a) and micro-window #2 in (b). The error bars represent two standard deviations of the fitting errors. The corresponding histograms of the fitting errors for all channels are shown on the right. For micro-window #1, the mean and standard deviation are -0.0027 K and 0.14 K, respectively. For micro-window #2, they are 0.0090 K and 0.15 K, respectively.

    * Did you also try to retrieve using the entire NH3 band? What were the results? Can both retrievals be combined to provide a combined column? If columns are simply averaged, what to do with the corresponding retrieval uncertainty and averaging kernel? A discussion is missing on this. In the NH3 dataset that is shared online, what are the variables that are included?

The retrievals were not done using the entire band. We expect the retrievals using a wider window may show difference that can be traced back to the sensitivity difference. In the main manuscript, we focused on the GIIRS $NH_3$ retrieval results using just the strong micro-window, 955-975 cm$^{-1}$, and moved the comparison with retrieval results using 920-940 cm$^{-1}$ micro-window to the supplementary materials.

In the file shared online, the variables include:
1) [Obs_Lat]: latitude of the observation
2) [Obs_Lon] : longitude of the observation
3) [Obs_Hour] : decimal hour (UTC) of the observation

4)  [Obs_SZA] : solar zenith angle of the observation (degree)
5)  [Obs_VZA] : viewing zenith angle of the observation (degree)
6)  [AP_SurfSkinT]: a priori for the surface skin temperature (Kelvin)
7)  [AP_SurfP]: a priori for the surface pressure (hPa)
8)  [AP_SurfEmissivity]: a priori for the surface emissivity
9)  [AP_AirMidLayerPres]: atmospheric pressure at mid-layer (hPa)
10) [AP_AirMidlayerTemp]: atmospheric temperature at mid-layer (Kelvin)
11) [AP_NH3_ColumnProf]: a priori for NH3 partial column profile (molecules/cm$^2$)
12) [AP_NumRetLayer]: number of layers used in the retrieval algorithm (<=11 layers)
13) [AP_NumPresLayer]: number of layers in the RT model (19 layers)
14) [Ret_NH3_ColumnProf]: Retrieved NH3 partial column profile (molecules/cm$^2$)
15) [Ret_SurfT]: Retrieved surface skin temperature (Kelvin)
16) [Ret_AirMidlayerTemp_SF]: Retrieved scale factor for [AP_AirMidlayerTemp]
17) [Ret_RedChi2]: Reduced Chi square from the retrieval algorithm
18) [Ret_IfConverge]: Indicator of converge (1) or not (0)
19) [Ret_NH3_ErrorDiag]: The diagonal of retrieval error covariance matrix for NH3
20) [Ret_NH3_AverageKernel]: averaging kernel matrix for NH3
21) [Ret_NH3_FitRes_Rad]: spectral fitting error relative to the spectral continuum
22) [Ret_NH3_FitRes_BT]: spectral fitting error in brightness temperature (Kelvin)

We have added a data user guide in the shared file online

*Are there differences in retrieved profiles? Could you again for selected region(s) show the average retrieved NH3 profile (+ a priori?) with both windows? Alternatively, can you show the averaged scaling factor profile (retrieved NH3 profiles divided by their a priori).

The following figure shows examples of the retrieved scale factors at different layers for (top) daytime (10h to 14h BJT) data in the North China Plain in July 2022; and (bottom) nighttime (22h to 2h next day BJT) data in North India in December 2022. We can see that the retrieved scale factor profiles agree well between the two micro-windows.

[Figure]

**Figure. Examples of the retrieved scale factors at different layers for (left) daytime (10h to 14h BJT) data in the North China Plain in July 2022; and (right) nighttime (22h to 2h next day BJT) data in North India in December 2022.**

*Figure 10: Can you add thermal contrast to this figure? (compare e.g. Fig. 4 of Clarisse et al., 2021). This would allow discussing better the variable sensitivity as a function of time.

You could also add the mean or median DOFS (unfiltered, total average).

In the revised manuscript, we have added Figure A2. It is an associated figure for Figure 10 on the diurnal change of TC and DOFS for the North China Plain and North India. Different from Figure 10, no extra filters have been applied.

 - Section 5

   * I would call this rather "Retrieval experiments".

Changed as suggested. In the revised manuscript, we have moved this to Section 3.

   * Did you add synthetic noise? (ideally generated randomly from the noise covariance matrix)

Yes. As explained in the text, the assumed noise according to the spectra noise of FY-4B/GIIRS is added to the simulated spectra.

   * I found section 5.2 not that interesting. Much rather it would be nice (see also comment above) to show and discuss the retrieved NH3 profiles along side temperature profiles and averaging kernels. This could be done with averages, but also on individual observations, showing cases e.g. of extreme temperature inversion (which forces the profile to a narrow band), very large thermal contrast, etc.. The paper does not nearly go deep enough on this aspect.

The original Section 5.2 has been moved into the supplemental materials. Following your suggestion, we added Figure A1 to show examples of the a priori and retrieval $NH_3$ profiles and the corresponding AK row vectors for three cases: (a) positive TC with TC=13.37K and DOFS=0.89 from daytime measurement on July 07, 2022; (b) negative TC with TC=-6.7K and DOFS=0.91 from nighttime measurement on December 18, 2022; and (c) weak TC with TC=3.39K and DOFS=0.23 from early evening measurement on July 06, 2022.

The following statements have been added to Section 4.1:
**"Measurement sensitivity of $NH_3$ is driven by TC and the $NH_3$ abundance (Clarisse et al., 2010). This is illustrated in the Appendix Figure A1(a) and (b) for a large positive and negative TC. As described above, while a positive TC leads to stronger absorption features, a negative TC causes spectral emission features allowing the detection of $NH_3$ also during the night (see also the example GIIRS spectra shown in Clarisse et al. (2021)). In both cases we see that the averaging kernels peak at the surface and the posteriori uncertainty in the retrievals of the surface layer are largely reduced compared to the a priori uncertainties. However, when the TC is small, as in the Appendix Figure A1(c), the DOFS values become smaller, and the AVK peaks higher up in the atmosphere (in this case, in the second layer). The retrieved value remains close to the a priori and the posteriori error is almost the same as the a priori, indicating low information content of the measurement. These examples illustrate the importance of TC for infrared sounding of boundary layer $NH_3$. An important advantage of GEO compared to LEO IR sounders is that they make observations throughout the day, such that optimal measurement conditions (large TC) can be found more readily. The diel variations of TC and DOFS are illustrated in the Appendix Figure A2 for North China Plain and North India.  LEO IR sounders like IASI with an equator crossing times**

**at 9:30 am and 9:30 pm LT in general do not measure at the time where measurement sensitivity (or DOFS) is largest. The optimal time is found around noon."**

B. Minor comments

- Please detail exactly how thermal contrast was calculated (because "lower atmosphere") can mean a lot of different things. Also, is surface temperature used or the brightness temperature of the surface (the difference being the surface emissivity)

We have changed "lower atmosphere" to "the lowest atmospheric layer". The surface skin temperature is the physical temperature (in Kelvin) of the surface skin.

- Figure 1c: x-label should be wavenumber, not frequency

Changed as suggested.

- Figure 2: y-axes labels inconsistent (diff vs difference)

Changed as suggested. In the revised manuscript, this figure has been moved to Figure S4.

- Figure 2: I would not show a priori x1 or x2, it doesn't contribute or provide any new information. In this way, the second row can be removed altogether, since this difference is shown in the bottom panel. Also please refer to the appropriate section for the definition of a priori (which is not introduced yet at this stage of the paper).

Thank you for your suggestion. We have removed the results with a prior x1 or x2. We added "**based on the a priori NH3 profiles shown in Figure 3**" in the figure caption to link to the a priori profile.

- Line 159: How was the measurement error covariance matrix determined?

The following statement has been added to define the measurement error covariance matrix:
"**$S_\varepsilon$ is the measurement error covariance matrix, which is assumed to be a diagonal matrix constructed using the spectra noise estimates**"

- Line 170: this is technically incorrect. It all depends on the magnitude of Sa. If the retrieval is poorly constrained (very large Sa), then the DOFS will naturally be high, as all info comes from the measurement rather than the a priori. Conversely, if the retrieval is very (too) tightly constrained, the DOFS will always be small.

We have rephrased this statement to be:
"**For example, a DOFS of 1.0 means that, given the assumed $S_a$, at least one independent piece of information can be retrieved from the spectral measurement to constrain the vertical distribution of NH₃. Note that the DOFS is highly dependent on the magnitude of the assume $S_a$, an indicator of the a priori knowledge. If $S_a$ characterizes a weaker constraint, indicating less a priori knowledge, the DOFS will be higher as relative more information will be taken from the measurement.**"

- Line 191: I guess this should read satellite zenith angles instead of solar zenith angles?

Changed as suggested.

- Line 234: this is factually incorrect, one can have a lower thermal contrast with increasing surface temperature. TC is a temperature difference, and for instance (Tsurf=290, Tair= 280 K) has a higher thermal contrast then (Tsurf = 300 K, Tair = 299 K). In fact the entire passage lines 233 - 237 need to be rewritten. I suggest the authors consult again the papers they reference just above, for proper terminology and physical explanations.

We have rephrased the statement to be:
**"When thermal contrast is close to zero, measurement sensitivity is low, and DOFS are close to zero. Large positive TC increases sensitivity and results in $NH_3$ spectral signatures that are seen in absorption. Large negative TC also allows for sensitive measurements, this time allowing $NH_3$ spectral signatures to be seen in emission. Negative TC corresponds to the situation where the atmosphere is warmer than the surface, allowing to decorrelate the surface layer with the rest of the lower troposphere."**

- What is the spectral resolution of the emissivity atlas that was used. Please mention this in the manuscript.

We have added the following statement in Section 3.1:
**"The emissivity values at 925 $cm^{-1}$ and 1075 $cm^{-1}$ are used to estimate the a priori emissivity for the retrieval micro-window. Two factors (slope and curvature) are used in the state vector to scale the wavelength dependent emissivity values"**

C. English. Although in general not bad, there are several typos/grammar mistakes. I definitely did not try to be exhaustive, so here are just some that I noted:

We have changed the following. At the same time, our co-authors (including L. Clarisse and M. Van Damme) have carefully proofread the paper.

- Line 15 and 75: to imply > to quantify / study / analyse (?)
Changed to "to quantify"
- Line 310 availabel
Changed to "available"
- Line 77: remaining > remainder of (?)
Changed to "the remainder of"
- Line 83: inconsistency east Asia vs East Asia
Changed to "East Asia"
- Line 100: constraints (?)  +  remove "measuring" + remove "column"
Changed to "makes it possible to accurately retrieve $NH_3$ over East Asia"
- Line 113: "can be referred to" > can be found in
Changed to "can be found in"
- Line 134: the NH3 > NH3
Changed.

Changed.

Changed to "Previous studies by **Clarisse et al. (2010, 2021)** and **Bauduin et al. (2017)** using IASI observations have shown that the DOFS is primarily driven by the TC."

Changed to "In $NH_3$ source regions (e.g., North China Plain and North India), the DOFSs are higher for the same TC compared with non-source region (e.g., Mongolia), suggesting the contribution from higher $NH_3$ concentration to the total information content."

Changed.

Changed.

---

## Author Response (AR2)

We thank the reviewer for his/her constructive comments and suggestions to improve the quality and clarity of our manuscript. We have made careful modifications to the manuscript according to all the comments and suggestions from the reviewer.

Item-by-item responses to the specific comments are provided below, in which the reviews' comments are in blue, our responses in **black**, and modifications of the original manuscript are indicated by highlight in yellow in the revised manuscript.

Review of Zeng et al. (2023)
Zeng et al. have considerably improved their manuscript according to the comments of the previous round of reviews. All of my comments from the previous review have been considered. In particular I want to acknowledge the efforts made to include the comparison the IASI data in the new section 4.3. From my point of view the manuscript is almost ready for publication, I only wanted to mention two minor points below, which could be addressed just before typesetting.

Specific points:
- Table 1: What does the column "no. of variables" mean? Are these the number of atmospheric layers in case of trace gases? For the interfering trace gases, it is stated in the text that a-priori profiles are scaled, so I would there only expect one "no. ov variables". Or do I understand this column wrong?

Yes, it is the number of variables in the state vector. For NH3 and H2O, they are 11 atmospheric layers. For the interference gas, it is 1 for each gas. In total there are 5 variables for the five interference gases.

In Table 1, we changed "5" to "**5 (in total)**"

- Section 4.3: I really like the new comparison to IASI now, thanks a lot for this! The only point within this comparison that I am missing is a discussion of the agreement between the data sets including the estimated errors (which has been done in the the previous sections). I think this could be simply done by adding a statement that the data agrees (or not agrees) within the estimated errors of both instruments.

We added "**Except for the daytime data in summer (July and August), both retrievals agree within the estimated error as quantified by rmse in Section 3.5**"